## ARTICLES
# Genomic analysis defines clonal relationships of ductal carcinoma in situ and recurrent invasive breast cancer

Esther H. Lips [1,24], Tapsi Kumar [2,3,4,24], Anargyros Megalios [5,24], Lindy L. Visser [1], Michael Sheinman[6], Angelo Fortunato[7,8], Vandna Shah[5], Marlous Hoogstraat [6], Emi Sei[3], Diego Mallo [7,8], Maria Roman-Escorza [5], Ahmed A. Ahmed [5], Mingchu Xu[2], Alexandra W. van den Belt-Dusebout[1], Wim Brugman[9], Anna K. Casasent[3], Karen Clements [10], Helen R. Davies[11], Liping Fu[1], Anita Grigoriadis [5], Timothy M. Hardman[12], Lorraine M. King[12], Marielle Krete[9], Petra Kristel[1], Michiel de Maaker[1], Carlo C. Maley[8], Jeffrey R. Marks[12], Brian A. Menegaz [13], Lennart Mulder[1], Frank Nieboer[1], Salpie Nowinski[5], Sarah Pinder [5], Jelmar Quist[5], Carolina Salinas-Souza[5], Michael Schaapveld[14], Marjanka K. Schmidt[1], Abeer M. Shaaban [15], Rana Shami[5], Mathini Sridharan[5], John Zhang[2], Hilary Stobart[16], Deborah Collyar [17], Serena Nik-Zainal [11], Lodewyk F. A. Wessels [6,18], E. Shelley Hwang[12], Nicholas E. Navin [3], P. Andrew Futreal [2], Grand Challenge PRECISION consortium*, Alastair M. Thompson [13,25], Jelle Wesseling [1,19,20,25] and Elinor J. Sawyer [5,25] ✉

**Ductal carcinoma in situ (DCIS) is the most common form of preinvasive breast cancer and, despite treatment, a small fraction (5–10%) of DCIS patients develop subsequent invasive disease. A fundamental biologic question is whether the invasive disease arises from tumor cells in the initial DCIS or represents new unrelated disease. To address this question, we performed genomic analyses on the initial DCIS lesion and paired invasive recurrent tumors in 95 patients together with single-cell DNA sequencing in a subset of cases. Our data show that in 75% of cases the invasive recurrence was clonally related to the initial DCIS, suggesting that tumor cells were not eliminated during the initial treatment. Surprisingly, however, 18% were clonally unrelated to the DCIS, representing new independent lineages and 7% of cases were ambiguous. This knowledge is essential for accurate risk evaluation of DCIS, treatment de-escalation strategies and the identification of predictive biomarkers.**

DCIS is often detected during routine mammography, representing 20% of all screened breast cancers[1]. There is evidence that many DCIS lesions, particularly low-grade lesions, will never progress to invasive disease[2,3]. However, almost all patients with DCIS are still treated with surgery with or without radiotherapy to prevent progression to invasive disease that can occur years or even decades after the initial DCIS. Despite treatment, there remains a subgroup of patients who still develop recurrent disease as demonstrated in a study of 7,934 patients treated by lumpectomy with or without radiotherapy, where 5.3% developed an invasive recurrence and 3.8% a DCIS recurrence with a median follow up of 9.4 years[4].

[1]Division of Molecular Pathology, The Netherlands Cancer Institute, Amsterdam, The Netherlands. [2]Department of Genomic Medicine, The University of Texas MD Anderson Cancer Center, Houston, TX, USA. [3]Department of Genetics, The University of Texas MD Anderson Cancer Center, Houston, TX, USA. [4]MD Anderson UTHealth Graduate School of Biomedical Sciences, Houston, TX, USA. [5]School of Cancer and Pharmaceutical Sciences, Faculty of Life Sciences and Medicine, Guy's Cancer Centre, King's College London, London, UK. [6]Division of Molecular Carcinogenesis, Oncode Institute and The Netherlands Cancer Institute, Amsterdam, The Netherlands. [7]School of Life Sciences, Arizona State University, Tempe, AZ, USA. [8]Biodesign Center for Biocomputing, Security and Society, Arizona State University, Tempe, AZ, USA. [9]Genomics Core Facility, The Netherlands Cancer Institute, Amsterdam, The Netherlands. [10]Screening Quality Assurance Service, Public Health England, London, UK. [11]Early Cancer Unit, Hutchison/MRC Research Centre and Academic Department of Medical Genetics, Cambridge Biomedical Research Campus, University of Cambridge, Cambridge, UK. [12]Department of Surgery, Duke University School of Medicine, Durham, NC, USA. [13]Department of Surgery, Dan L Duncan Comprehensive Cancer Center, Baylor College of Medicine, Houston, TX, USA. [14]Division of Psychosocial research and Epidemiology, The Netherlands Cancer Institute, Amsterdam, The Netherlands. [15]Queen Elizabeth Hospital Birmingham and University of Birmingham, Birmingham, UK. [16]Independent Cancer Patients' Voice, London, UK. [17]Patient Advocates in Research, Danville, CA, USA. [18]Faculty of Electrical Engineering, Mathematics, and Computer Science, Delft University of Technology, Delft, The Netherlands. [19]Divisions of Diagnostic Oncology, The Netherlands Cancer Institute, Amsterdam, the Netherlands. [20]Department of Pathology, Leiden University Medical Center, Leiden, The Netherlands. [24]These authors contributed equally: Esther H. Lips, Tapsi Kumar, Anargyros Megalios. [25]These authors jointly supervised this work: Alastair M. Thompson, Jelle Wesseling, Elinor J. Sawyer. *A list of authors and their affiliations appears at the end of the paper. ✉e-mail: elinor.sawyer@kcl.ac.uk

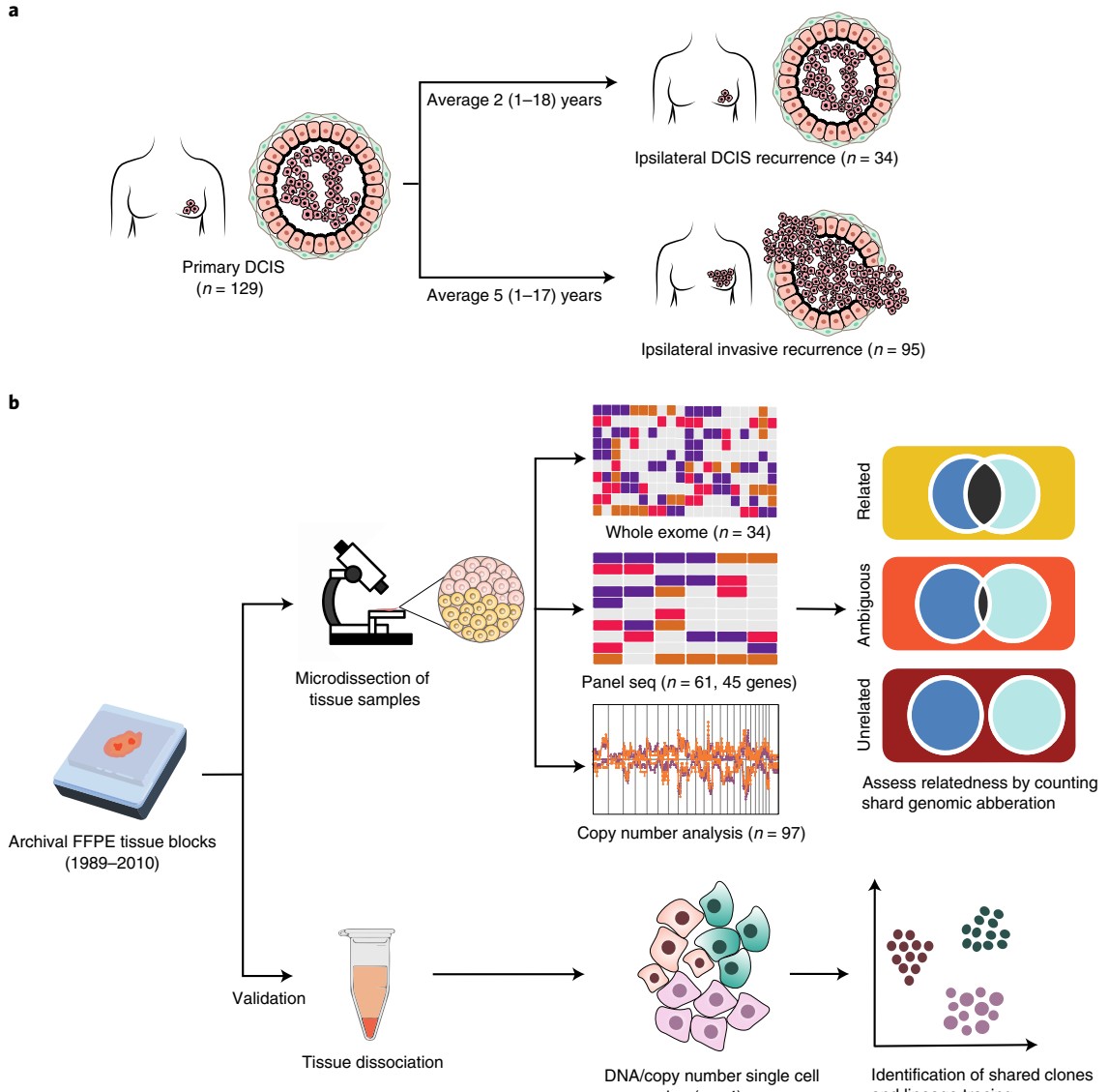

**Fig. 1 | Study design. a**, Graphical representation of our clinical cohort with long-term follow up to study clonal relatedness between primary DCIS and subsequent disease. The two different groups of subsequent recurrences (ipsilateral DCIS recurrences and ipsilateral invasive recurrences) are shown, together with sample numbers and the median time to follow up. **b**, The two different strategies undertaken to unravel clonality in DCIS with subsequent disease. First, we microdissected a large cohort of DCIS recurrence pairs and analyzed them with whole exome sequencing, panel sequencing and copy number analyses. Subsequently, we assessed clonal relatedness by counting the number of shared mutations and copy number aberrations. Second, as means of validation, tissue of paired lesions was dissociated, followed by scDNA-seq to study shared tumor subclones.

As many DCIS lesions will never progress to invasive disease, some women may receive intensive treatment without any clinical benefit[2,3]. Therefore, there is a great unmet clinical need to develop treatment strategies that avoid overtreatment. Clinical trials of treatment de-escalation are currently underway, for example, by leaving out radiotherapy or even by refraining from surgery in the absence of invasion[5,6]. However, there is limited knowledge of whether invasive recurrences are clonally related to the initial DCIS disease, making accurate evaluation of the risk of progression and the assessment of the prognostic value challenging. Detailed analysis of this question in a large cohort of patients with matched recurrent tissue samples and long-term clinical outcome data are currently lacking.

Thus, the central question of this study is whether the initial DCIS and subsequent 'recurrence' share a common genetic lineage or, alternatively, represent independent diseases that emerge from different initiating cells in the same breast (ipsilateral). This question

has been difficult to address, in part due to the logistical challenges in collecting matched longitudinal samples that are years to decades apart, and the technical challenges in performing genomic assays on archival formalin-fixed paraffin-embedded (FFPE) materials of this age. Consequently, most studies have focused on studying synchronous ductal carcinoma, which are single timepoint samples that have areas of DCIS and regions of invasive cancer cells co-occurring in the same tissue section. These studies represent more advanced cases, in which invasion has already occurred, and as expected most data have shown the cancer cells from the in situ and invasive areas are clonally related in their genetic profiles[7-10].

In contrast, the genomic data on 'pure DCIS' with matched recurrent DCIS or invasive tumors from many years later is far more limited. One small study, that used genomic data to assess clonal relatedness, showed that in two of eight pairs of primary DCIS and subsequent invasive disease copy number aberrations were not concordant[11],

suggesting that perhaps not all cases are clonally related to the initial DCIS. Thus, while the subsequent cancers following DCIS are often termed 'recurrences' as a clinical definition, the genetic studies supporting their clonal relationship is still lacking.

Here, to investigate the genomic concordance and clonal relationship of pure DCIS and subsequent ipsilateral recurrent DCIS or invasive cancers, we pooled samples from three countries, resulting in the largest cohort to date of DCIS cases with 5–17 years of clinical data follow up (Supplementary Data Table 1). We applied genomic profiling methods to systematically and comprehensively investigate genomic concordance of pure DCIS and recurrent invasive tumors, using exome sequencing, targeted mutation panels and copy number profiling. We further applied single-cell DNA sequencing (scDNA-seq) methods to validate these results in a subset of cases ($N = 4$).

## Results

**Overview of clinical samples profiled.** In total, 129 primary DCIS and their matched recurrences were analyzed, of which 95 recurred as invasive breast cancer and 34 as a second DCIS (Fig. 1 and Supplementary Data Table 2). All recurrences were ipsilateral, meaning that they occurred in the same breast as the initial primary DCIS lesion. The median age at diagnosis of the primary DCIS was 57 years (range 34–87 years) and median time to the recurrence was 4 years (0.4–17.5 years). Of the primary DCIS samples, 52% were high grade, 67% were estrogen receptor (ER) positive (ER+) and 29% were HER2 receptor positive (Supplementary Data Table 3). Only 13% (12/95) of the primary DCIS that developed an invasive recurrence received radiotherapy as part of their primary treatment, in contrast to 53% (18/34) of the primary DCIS that recurred as pure DCIS.

**Whole exome sequencing to assess clonal relatedness.** We employed whole exome sequencing (WES) to survey somatic mutations in 24 DCIS-invasive (INV) recurrence pairs. The numbers of shared and private mutations were highly variable for the different tumor pairs, ranging from 0 to 112 shared mutations and from 4 to 646 private mutations (Fig. 2a–c and Supplementary Data Table 4). Details of all mutations detected can be found in Supplementary Data File 1. Shared mutations had significantly higher ($P < 0.001$, Wilcoxon Rank Test) allele frequencies compared with private mutations (Fig. 2d), consistent with early clonal selection, with the most common shared mutations occurring in *TP53* and *PIK3CA* (Fig. 2b–c and Extended Data Fig. 1a). Invasive recurrences had a higher number of private mutations than their matched primary DCIS, ($P = 0.039$, Wilcoxon Rank Test, two-sided), (Extended Data Fig. 1b).

Clonal relatedness was assessed using Breakclone, a statistical approach we developed (Methods) that, in contrast to other existing algorithms[12–14], incorporates both population frequency and allele frequency when using mutation data, and the position of the individual copy number aberration breakpoints when using copy number data. Taking into account the population frequency of different aberrations serves to give greater weight to true clonal events, while down-weighting aberrations characteristic of certain cancer types that are more likely to recur independently across different tumors.

Breakclone computes a clonal relatedness score and a *P* value based on a permutation test and designates a pair as being either related ($P < 0.05$), ambiguous ($0.05 < P < 0.1$) or unrelated ($P > 0.1$) (Fig. 2a).

Of the 24 DCIS-INV cancer pairs, 83% (20/24) showed clear evidence of clonal relatedness, including three cases of primary DCIS that developed an invasive recurrence despite having undergone a mastectomy (Supplementary Data Table 3). The remaining four pairs (17%) did not harbor any shared mutations that could be detected in our analyses, with 11–70 mutations being detected in the primary DCIS and 36–329 in the invasive recurrence (Fig. 2a–c and Supplementary Data Table 4). Lineage inference of clonally unrelated pairs showed clearly different subclones and drivers in the primary and recurrence lesions (Fig. 2e). Analysis of the clonally related pairs revealed that the primary DCIS consisted of multiple subclones, some of which had expanded in the invasive recurrence, dominant subclones remained at high frequencies in both the primary and recurrent samples (Fig. 2f).

Our results based on WES show that most DCIS-INV recurrence pairs are clonally related, and further show that, in some cases, the genomic profiles are highly similar. Notably, however, others had diverged genomically, acquiring many additional events but are still related to the DCIS through a common ancestor. These data also revealed a small number of cases ($N = 4$) that did not share any genomic alterations between primary DCIS and invasive recurrences, suggesting that an independent tumor, representing a second cancer, emerged in the same breast.

**Copy number and mutation analysis to assess clonal relatedness.** We next analyzed an additional 71 DCIS-INV recurrent pairs by genomic copy number analysis using either single nucleotide polymorphism (SNP) array or low-pass whole-genome sequencing (lpWGS) data (Methods). Of the 62 cases that passed quality control (QC), 71% (44/62) were considered clonally related, 2% (1/62) ambiguous and 27% (17/62) unrelated (Fig. 3a and Extended Data Fig. 2). Examples of individual copy number profiles of related and unrelated pairs are shown in Fig. 3b and Supplementary Fig. 1.

In 45 of the 71 pairs that underwent copy number analysis there was sufficient DNA to also perform targeted sequencing (Supplementary Data File 1), which revealed that 51% (23/45) were classified as clonally related (including four considered unrelated by copy number) and 15% (7/45) unrelated (all supported by copy number data; Supplementary Data Table 3). A further 33% (15/45) were classified as ambiguous, with only a single mutation identified and shared between both matched samples. In 11 of these cases, copy number data confirmed clonal relatedness.

**Combined results classify 75% of invasive recurrences as clonal.** We combined our classifications based on WES, PanelSeq and copy number data to obtain a final call on the clonal relatedness for each patient. In cases of conflicting data between the different platforms, the clonal relatedness classification prevailed over unrelatedness (Fig. 4a). There were nine pairs that had both targeted sequencing data and WES; in seven of these, the results were concordant, and in two a single mutation was found to be shared on targeted sequencing but not detected on WES. Manual inspection of the WES data revealed that these mutations were present but had not

**Fig. 2 | Clonality assessment using whole exome sequencing. a**, The total number of mutations, followed by the mutations type (primary private, recurrence private and shared), the breakclone score and the final clonality conclusion plotted for each DCIS-invasive recurrence pair. **b**, Scatter plots showing the VAF of mutations in three clonal related pairs. **c**, As in **b**, for three clonal unrelated pairs. **d**, Boxplots comparing VAF primary private ($n = 702$), recurrence private ($n = 1,257$) and shared mutations ($n = 433$). Minima and maxima are present in the lower and upper bounds of the boxplot, respectively. Adjusted *P* values for Holm–Bonferroni method $P = 5.96 \times 10^{-84}$ and $P = 3.84 \times 10^{-39}$ were calculated with two-sided Wilcoxon test. For the shared mutations, both the VAF in the DCIS tissue (Primary shared) and the invasive recurrence (Recurrence shared) are shown. Center solid lines represent the median, box edges show the 25th and 75th percentiles and whiskers represent the maximum and minimum data points within 1.5× the interquartile range outside box edges. **e**, Lineage tracing for two patients with clonal unrelated tumors. **f**, As in **e**, for two clonal related tumor pairs.

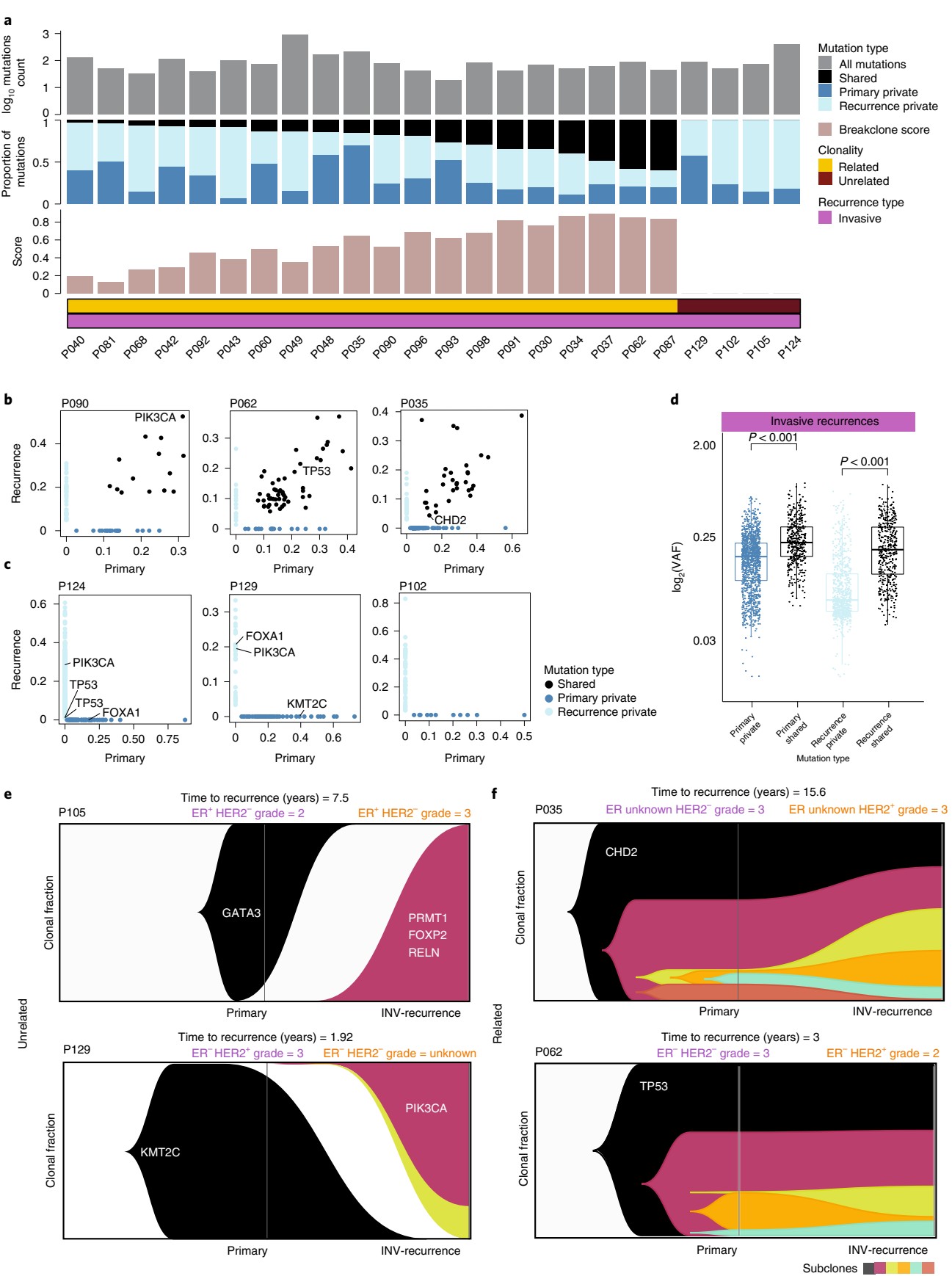

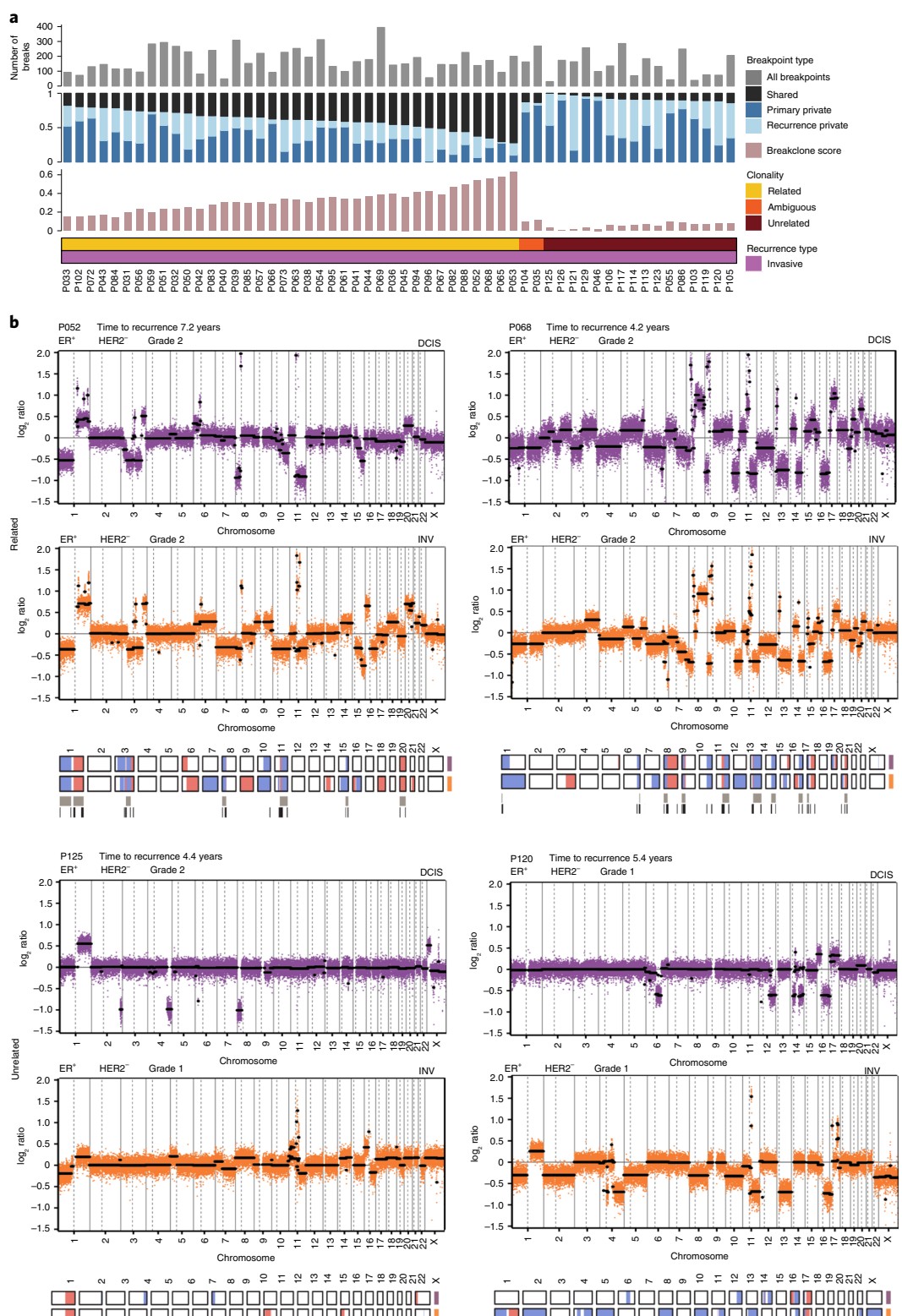

**Fig. 3 | Validation of clonality using copy number profiling. a**, Distribution of breakpoints in primary DCIS and recurrent invasive pairs derived from copy number in lpWGS. The top row (gray) shows the total number of breakpoints for each patient, the next row whether the breakpoints were unique (private) to the primary or recurrence or shared and the final row (pink) the breakclone score. **b**, Genome-wide segmented copy number profiles and called aberrations heatmaps of two clonally related (P052, P068) and two clonally unrelated (P125, P120) pairs, illustrating relatedness between primary DCIS (purple) and its paired recurrent invasive disease (orange) based on lpWGS copy number analysis. In the copy number profile plot, raw log ratios are in color and segmented log ratios are in black. Called aberrations of gains (red) and losses (blue) are presented in heatmaps below. Shared aberration events (top bar; gray) and shared breakpoints (bottom bar, black) between pairs are shown underneath the heatmaps. The genomic position is indicated by chromosome 1 on the left and up to chromosome X on the right in both graphs.

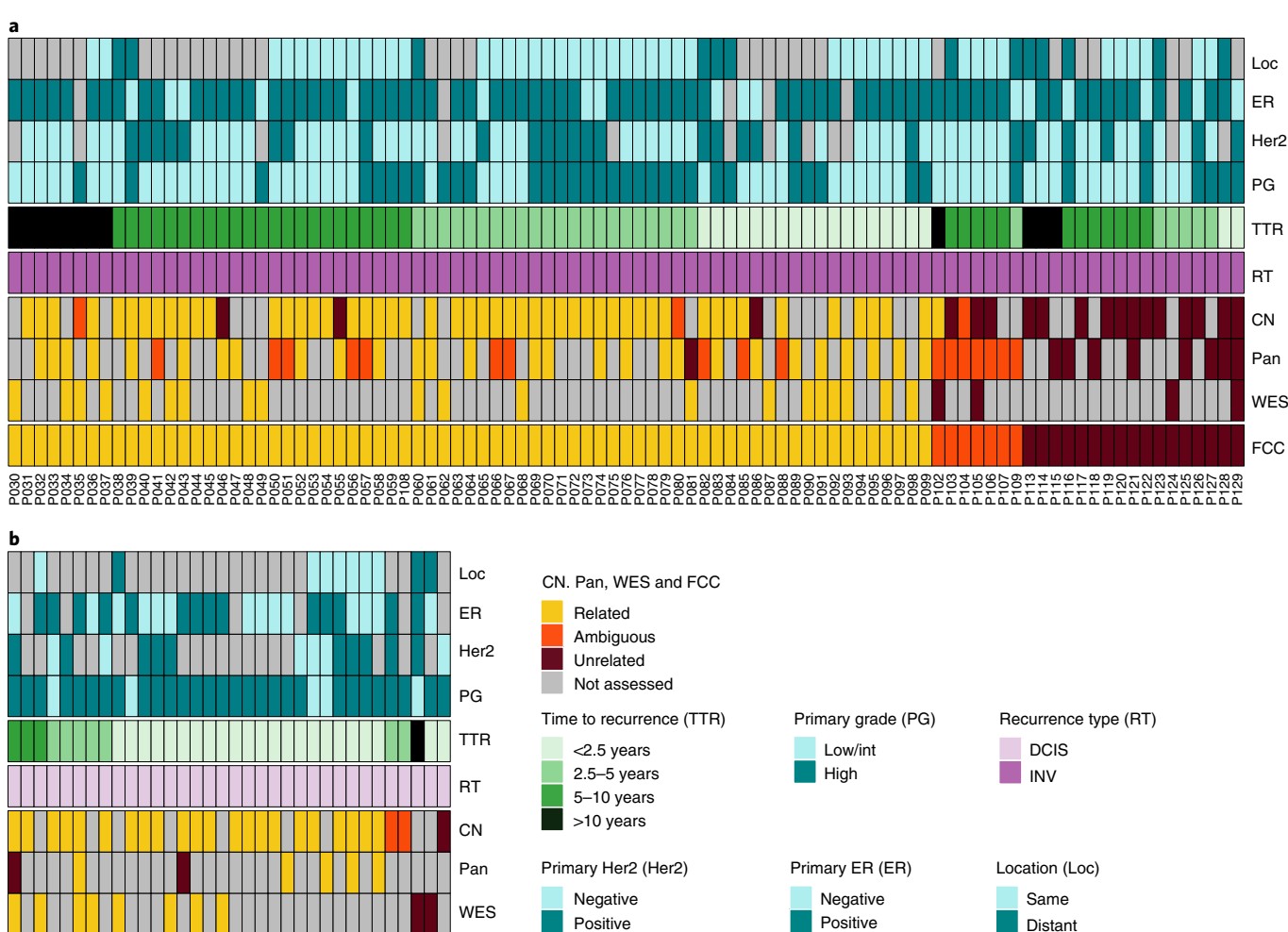

**Fig. 4 | Summary heatmap of clonality calls and clinical characteristics.** The clonality calls calculated by copy number (CN, panel sequencing (Pan) and WES are shown as well as the final clonality call (FCC). The samples are ordered by subsequent FFC, time to recurrence (TTR) and location (Loc) overlap between primary DCIS and recurrence. **a**, Summary of primary DCIS-invasive recurrence pairs. **b**, Summary of primary DCIS-subsequent DCIS recurrence pairs.

passed the QC thresholds due to the high variant allele frequencies of the mutations in the paired normal, probably due to tumor contamination of adjacent normal tissue. In summary, when data were aggregated on all samples of DCIS-INV recurrence pairs across all three analysis platforms (WES, panel sequencing and copy number analysis), 75% (71/95) DCIS-INV pairs were classified as clonally related, 18% (17/95) were unrelated and 7% (7/95) were ambiguous (Fig. 4a and Supplementary Data Table 3).

**DCIS–DCIS recurrence pairs are more frequently clonal.** Not all recurrences following DCIS are invasive; in the first 5 years, pure DCIS recurrence is almost as common as invasive recurrence and then decreases, whereas the risk of invasive recurrence increases consistently over 10 years[4]. We therefore analyzed 34 pairs that recurred as pure DCIS (Supplementary Data Table 5), 9 by WES (Extended Data Fig. 3a–d) and 25 by copy number profiling with or without additional targeted sequencing (Extended Data Fig. 3f). Of these 34, 85% (29/34) were found to be related, 9% (3/34) unrelated and 6% (2/34) ambiguous, suggesting that the pure DCIS recurrences are likely to be residual DCIS that was not detected by conventional imaging preoperatively and remained in situ following

surgery (Fig. 4b). DCIS cases that recurred as pure DCIS tended to recur earlier than those that developed an invasive component (mean 36 versus mean 65 months, respectively, $P = 0.0003$, $t$-test, two-sided). However, there were also late clonal pure DCIS recurrences, (P003) which relapsed with pure DCIS after 8 years despite radiotherapy and endocrine therapy; both primary and recurrence shared a pathogenic *TP53* driver mutation (Extended Data Fig. 3d).

**scDNA-seq to reconstruct clonal lineages.** To validate the bulk genomic profiling classifications, we applied scDNA-seq to profile genomic copy number in 2,294 cells from primary and recurrent disease from FFPE tissue collected from four DCIS patients (Methods). In the two clonally unrelated sample pairs (P122 recurred as invasive disease, P110 as pure DCIS), unbiased clustering identified three main subclones in P122 and seven subclones in P110, in which each of the individual clones was specific to either the primary DCIS or recurrent tumors (Fig. 5a). In P122, the clustered heatmaps showed that subclone 1 (c1) was specific to the primary DCIS and had a number of copy number aberration (CNA) events but showed no common CNA events or breakpoints with the recurrent subclones (c2–3) (Fig. 5b). Similarly, in the nonclonal

P110 patient, common chromosomal losses on 16q and 17p (*TP53*) were detected in all of the subclones (c1–5) from the primary sample but did not share any CNA events with the subclones (c6–c7) in the invasive disease. We computed consensus subclone CNA profiles from the single cells (Extended Data Fig. 4c) and reconstructed clonal lineages (Extended Data Fig. 4d) and Muller plots of subclonal frequencies[15], which confirmed the independent lineages in both DCIS tumors (Fig. 5c). We further investigated the clonal substructure of two clonally related patients (P082 and P042) classified by bulk DNA-seq. Unbiased clustering identified eight subclones in P082 and six subclones in P042 (Fig. 5d). In contrast to the two clonally unrelated pairs, these tumors shared a large number of CNA events between the primary and invasive tumors (Extended Data Fig. 4a,b). In P082, chromosomal gains in 8q (*MYC*), 17q (*ERBB2*) and 20 (*AURKA*), and losses in 11q (*PGR*), 16q and 17p (*TP53*) were shared among all eight subclones, whereas in P042, chromosomal gains in 1q, 8q (MYC) and 17q (*ERBB2*), and losses in 8p, 11q(*PGR*) 16q and 17p(*TP53*) were present in all six subclones. Furthermore, in P082, multiple subclones (c1, c2, c3, c7, c8) with the same genotypes were detected in both primary and invasive disease. Consensus subclone CNA profiles were computed from single cells (Extended Data Fig. 4b) and used to reconstruct clonal lineages (Extended Data Fig. 4d), which identified subclones that expanded in the invasive disease and harbored CNA events associated with recurrence, including subclones c4–c7 in P082 and c4–c6 in P042 (Fig. 5e). Collectively, the single-cell data validated the clonal classifications estimated by bulk DNA-seq and further resolved direct and independent clonal lineages, revealing chromosomal events and genes associated with recurrence.

**Genomic aberrations in recurrent invasive disease.** In the patients classified as clonally related, we compared specific CNAs and mutations between the primary DCIS and invasive recurrences to identify genomic events that occurred at a higher frequency in the invasive disease and were thus associated with invasive recurrence. Strikingly, our data show that most mutations and CNAs detected in the matched invasive breast cancer were already established in the primary DCIS and there were no clear genomic markers of invasive progression. The genes most frequently mutated were *PIK3CA* (24% DCIS, 27% INV) and *TP53* (24% DCIS, 27% INV) (Fig. 6a) and the most common amplicons were on 17q12 (*ERBB2*: 29% DCIS, 27% INV), 17q21.1 (*GSDMB*, *PSMD3*: 29% DCIS, 25% INV) and 11q13 (*CCND1*: 20% DCIS, 20% INV) (Supplementary Data Table 6). A frequency analysis of CNA profiles across all DCIS and invasive recurrences showed highly similar chromosomal gains and losses across the patient cohort (Fig. 6b). However, 1q and 8p11 gain were more common in recurrent invasive disease (38% DCIS, 58% INV *P* = 0.03 and 18% DCIS, 38% INV, *P* = 0.01, respectively, Fishers exact test, two-sided) and 3p21 loss in primary DCIS (40% DCIS, 20% INV, *P* = 0.03, Fishers exact test, two-sided; Fig. 6b). The latter may represent a subclone present in the primary DCIS that did not progress into invasive disease. Although the fraction of the genome altered overall did not differ significantly between DCIS and recurrent invasive disease, regions of copy number gain were more common in recurrent invasive disease (Extended Data Fig. 5). Collectively, these data suggest that most copy number events and

driver mutations in invasive breast cancer have already occurred at the earliest stages of progression in DCIS, many years before the emergence of the invasive disease and were not associated with invasive recurrence.

**New invasive tumors can arise from independent DCIS lesions.** A large proportion (64%) of the invasive recurrences also had evidence of adjacent DCIS in the tissue sections. In ten cases there was enough recurrent adjacent DCIS to analyze the lesions separately and, in all cases, the recurrent DCIS was clonally related to the adjacent recurrent invasive disease (eight samples by copy number, two samples by WES; Supplementary Data Table 7). In eight cases, the recurrent DCIS and invasive disease were clonally related to the primary DCIS. In one such case (P098), WES revealed that the recurrent invasive disease was comprised of four subclones, two of which were detected in the initial primary DCIS and two that appeared in the recurrent synchronous DCIS and invasive disease (Extended Data Fig. 6). In another patient (P087) lineage tracing showed two main subclones in the recurrent invasive disease and associated DCIS, of which one was also present in the primary DCIS. In two cases, the recurrent DCIS and invasive disease were unrelated to the primary DCIS, indicating that the de novo primary invasive tumors arose from the new independent DCIS lesions (Supplementary Data Table 7).

**Clinical characteristics do not predict clonal relatedness.** To assess whether clinical features can be used to predict which pairs are most likely to be nonclonal, we tested for associations with standard clinical parameters (Supplementary Data Table 8). Nonclonal pairs were more likely to have discordant ER status (*P* = 0.003, Fisher's exact test, two-sided) and to occur distant from the site of the primary DCIS (*P* = 0.03, Fisher's exact test, two-sided). However, there were no significant associations between clonal relatedness and time to recurrence, age at diagnosis of primary DCIS, treatment with radiotherapy, ER/HER2 receptor status or grade of primary DCIS. These data show that relatedness between primary DCIS and a subsequent recurrence cannot be assessed with high accuracy based on clinical data alone.

**Estimation of de novo invasive primary rate following DCIS.** We calculated age and period standardized incidence ratios (SIRs) to estimate the risks of ipsilateral and contralateral new primary invasive breast cancer in women with DCIS who underwent wide local excision (WLE) with or without radiotherapy compared with the general population using the Dutch cancer registry data (https://iknl.nl/nkr). Based on our clonality results, we assumed that 18% of ipsilateral recurrences and 100% contralateral recurrences following DCIS were new primaries. The latter assumption is supported by the clonal analysis of the 34 contralateral recurrences used to validate the Breakclone algorithm and presented in Methods. The SIR was 2.10 (95% confidence interval, 1.83–2.42) for women with DCIS who underwent WLE only, and 1.85 (95% confidence interval, 1.57–2.17) for women with DCIS who underwent WLE with radiotherapy. These results indicate that the de novo primary rate of invasive breast cancer following DCIS is significantly higher than in the general population, implying DCIS is also a risk factor for developing subsequent invasive cancer as well as a precursor lesion.

**Fig. 5 | Clonal lineage reconstruction by single-cell genome sequencing. a**, UMAP plots of single-cell copy number profiles from FFPE tissue showing clusters of subclones at primary timepoint or recurrence for two DCIS patients with independent lineages. **b**, Clustered heatmaps of single-cell copy number profiles for two DCIS cases where the recurrence event represents an independent lineage, with selected breast cancer genes annotated below the heatmap. **c**, Muller plots showing clonal frequencies and lineages reconstructed from neighbor-joining trees using timescape, with selected breast cancer genes annotated, and chromosomal gains and losses indicated by plus and minus signs, respectively. **d**, UMAP plots of single-cell copy number profiles from FFPE tissue for two clonally related pairs showing subclones at the primary DCIS and at the recurrence time points. **e**, Muller plots of the same two clonally related pairs showing clonal frequencies and lineages reconstructed from neighbor-joining trees using timescape, with selected breast cancer genes annotated, again with gains and losses annotated with plus and minus signs, respectively.

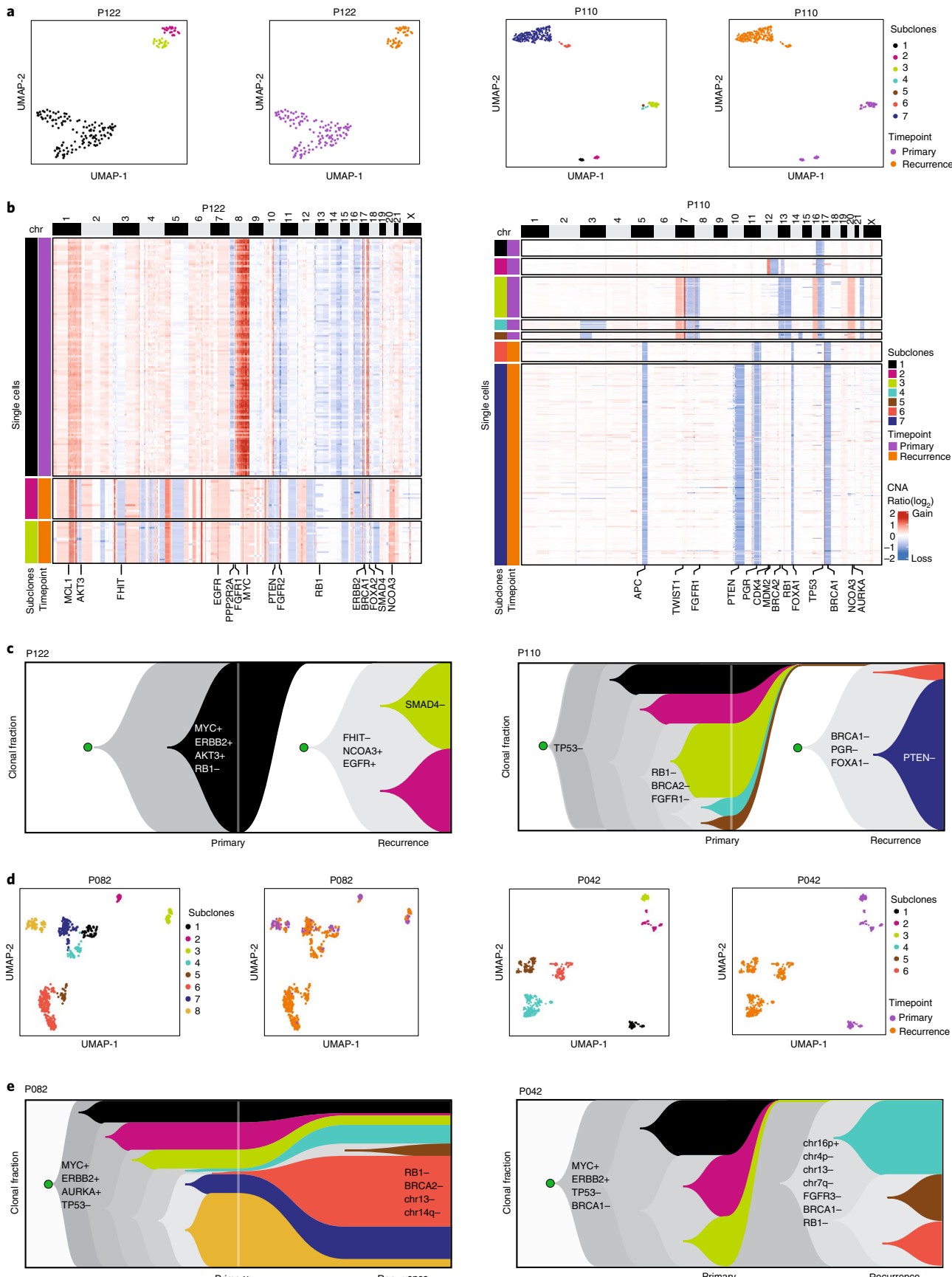

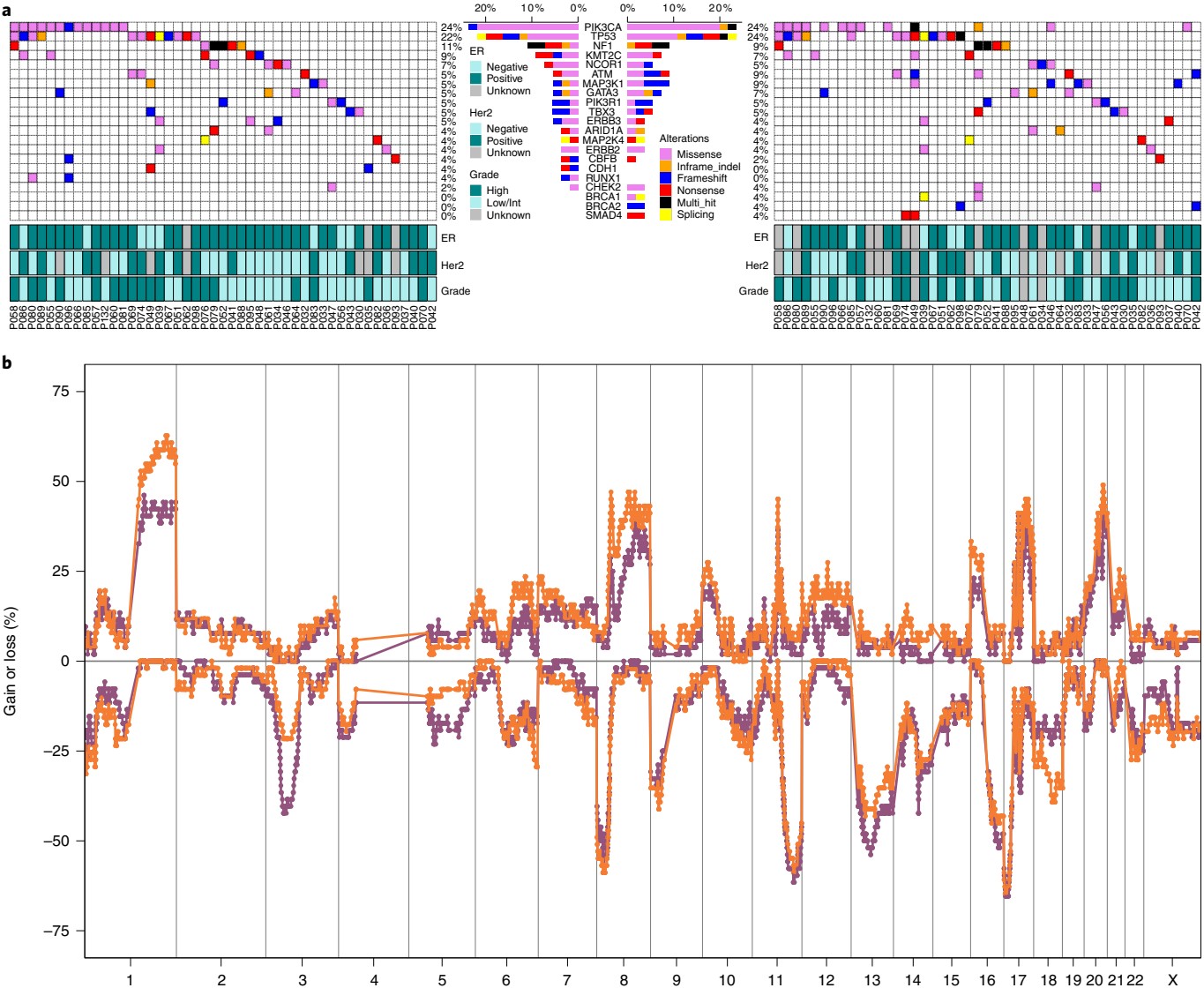

**Fig. 6 | Mutations and copy number alterations in primary DCIS and subsequent clonally related invasive recurrences. a**, Oncoplots for primary DCIS samples (left) and invasive recurrences (right) based on WES and targeted sequencing. Of the 45 genes covered by all sequencing platforms, only genes mutated in more than 3% of the primary DCIS or invasive recurrence samples are shown. We removed C>T mutations with allele frequency < 0.1 and fewer than three entries in the COSMIC database. **b**, Frequency plot of genome-wide copy number alterations in clonally related DCIS and invasive recurrences ($n = 55$) showing primary DCIS (purple) and its paired ipsilateral invasive recurrence (orange). The $y$ axis shows the percentage of samples with gains (above zero line) and losses (below zero line). The genomic position is indicated by chromosome 1 on the left and up to chromosome X on the right with chromosome boundaries indicated by vertical lines.

**Presence of pathogenic germline variants.** Although the study was not designed to investigate the presence of pathogenic germline variants, we were able to look for known pathogenic germline mutations (as defined by Clinvar) in *BRCA1, BRCA2, CHEK2, PALB2, TP53, CHEK2* and *ATM* in those cases where paired normal FFPE tissue had been sequenced (24 cases by WES, 16 by PanelSeq). The only variant detected in these normal FFPE tissues was a *PALB2* variant of which pathogenicity is uncertain (Supplementary Table 9).

In the 38 samples that underwent Panelseq without paired normal tissue we also looked for potential germline mutations in the above seven genes. However to minimize the likelihood of including any somatic mutations due to the lack of paired normal, we only included variants that have been described previously as germline pathogenic variants in clinvar and where the variant allele frequency (VAF) was >30% in both the DCIS and paired recurrence. Two variants were identified that are known pathogenic germline *BRCA2*

mutations, one of these has been described previously as somatic as well as germline so we cannot be certain it is a germline mutation in our sample. In both of these cases, the invasive recurrences were clonally related to the primary DCIS. Another four variants of unknown significance or conflicting pathogenicity were identified that have not been described previously as somatic; in two, the invasive recurrences were clonally related, in one equivocal and in the other unrelated (Supplementary Data Table 9). All potential germline variants were excluded from the clonality analysis.

## Discussion

In this large dataset of matched DCIS-INV recurrence pairs, we have confirmed that primary DCIS can be a precursor to subsequent invasive cancer. However, our data show that not all ipsilateral invasive breast cancers are clonally related to previous DCIS, but rather 18% represent de novo primary cancers in the same breast. As such,

our results confirm, on a much larger and detailed scale, the small study by Gorringe et al. showing unrelated recurrences[11]. In patients with nonclonally related recurrences, the major question is what factors constituted the basis of risk in these women who developed two clonally different cancers in the same breast over time. We can speculate that such cases may be due to genetic predisposition as there is clear evidence that the known invasive breast cancer predisposition genes and polymorphisms also predispose to DCIS, particularly ER+ DCIS[16,17]. However, the limited data on rare germline variants in this study does not support this theory. Another possibility is that a cancer field effect in the breast has greatly increased the probability of developing new cancers compared with the general population. The 'sick lobe' theory and field cancerization concepts may explain why, in an affected breast prone to tumorigenesis, despite wide local excision of DCIS with histologically clear margins and radiotherapy, a secondary tumor can emerge that is unrelated to the initial DCIS[18,19]. Studies on tumor-adjacent normal tissue raise the possibility that nonmalignant precancerous cells contribute to recurrences[20,21].

Our data also show that at least 75% of the invasive recurrences are clonally related to the initial DCIS diagnosis, sharing a common genomic lineage that was established from the same ancestral cell in the breast (Extended Data Fig. 7). The genomic data show a very high concordance in both the driver mutations and chromosomal amplifications that were detected in both the DCIS and invasive disease in these patients. These findings support that genomic evolution occurs at the earliest stages of breast cancer progression (within the ducts) in which driver events including *TP53*, *PIK3CA* mutations and HER2 amplifications are present at the DCIS stage, before breaking through the basement membrane of the ducts to establish the invasive disease. These findings suggest that specific genomic mutations per se do not drive invasion, but that perhaps a critical combination of mutations and CNAs is required, or, alternatively, that characteristics of the tumor and surrounding microenvironment are present at the earliest stages of progression to permit invasion in the later stages of the disease[9,22]. Further characterization of the DCIS microenvironment may reveal key stromal and immune factors that may create conditions that are permissive for invasion.

The finding that one in five ipsilateral invasive cancers following DCIS are not clonally related has fundamental biologic implications: first, DCIS can no longer be considered solely as a precursor lesion, but rather also a risk lesion for development of further invasive disease. This is similar to the role that has been ascribed to lobular carcinoma in situ (LCIS), where there is both an increased risk of subsequent ipsilateral and contralateral invasive disease[23]. Second, the true risk of recurrence from the same population of preinvasive tumor cells has probably been overestimated, thereby confounding the potential benefit of radiotherapy, as radiation probably prevents clonal progression rather than preventing initiation of a new neoplastic process. Third, these data have important implications for accurate identification of predictive biomarkers for invasive progression, since, in clonally unrelated DCIS, the notion of biomarkers predictive of invasion is irrelevant. These data may explain why it has been so challenging to identify predictive biomarkers of progression to invasive disease to date[24,25], further underscoring the need to characterize DCIS more comprehensively in the context of the stroma in future studies.

Important future directions will include identifying those factors that contribute to dormancy of DCIS cells and their reactivation to establish invasive disease years to decades later, and understanding the role of nongenetic factors, such as the tumor microenvironment, in invasive progression. These biological insights are essential to enable well-informed DCIS treatment decisions that will help avoid overtreatment of low-risk DCIS that likely will never progress, while still providing appropriately aggressive treatment for high-risk DCIS with greatest invasive potential.

In conclusion, we performed extensive genomic characterization of the primary tumor and matched recurrence in one of the largest cohorts of patients treated for DCIS who subsequently developed an ipsilateral invasive cancer. Although the majority of subsequent invasive cancers were clonally related to the primary DCIS, a substantial subset was unrelated to the index DCIS. Our findings show that DCIS is not only a precursor to invasive cancer, but also a potential marker of a field effect for which surgery may have a limited role. These insights suggest an increasingly important rationale for developing nonsurgical approaches to effectively reduce increased breast cancer risk in these patients.

## Online content

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

## Grand Challenge PRECISION consortium

Jelle Wesseling[1,19,20,25], Alastair M. Thompson[13,25], Serena Nik-Zainal[11], Elinor J. Sawyer[5,25], Helen R. Davies[11], P. Andrew Futreal[2], Nicholas E. Navin[3], E. Shelley Hwang[12], Jos Jonkers[1], Jacco[1], Fariba Behbod[21], Esther H. Lips[1,24], Marjanka K. Schmidt[1], Lodewyk F. A. Wessels[6,18], Daniel Rea[15], Proteeti Bhattacharjee[1], Hilary Stobart[16], Deborah Collyar[17], Donna Pinto[22], Ellen Verschuur[23] and Marja van Oirsouw[23]

[21]Department of Pathology and Laboratory Medicine, University of Kansas Medical School, Kansas City, KS, USA. [22]DCIS411, San Diego, CA, USA. [23]Borstkanker Vereniging Nederland (Breast Cancer Association Netherlands), Utrecht, The Netherlands.

## Methods

This research complies with all relevant ethical regulations: The Sloane project, United Kingdom National Health Service Breast Screening Programme, was approved by the UK Health Research Authority (Ethical approval REF 08/S0703/147, 19/LO/0648); The Dutch DCIS cohort study, a Netherlands Cancer Registry (NCR; reference no 12.281), nationwide network and registry of histology and cytopathology in the Netherlands (PALGA; reference no. LZV990) approved by the Central Committee on Research Involving Human Subjects in the Netherlands and the Institutional Review Board (IRB) of the Netherlands Cancer Institute (CFMPB166, CFMPB393 and CFMPB688); The Duke Hospital cohort approved by Duke University Health System IRB, USA (Pro00054877, Pro00068646).

**Statistics and reproducibility.** No statistical method was used to predetermine sample size. The research question we addressed ('What percentage of primary DCIS is clonally related to invasive recurrences?') has never been addressed on a large scale before; therefore, we collected all available paired samples from three large cohorts.

Samples were excluded due to the following reasons:

- Only one sample of a pair was available,
- DNA quantity was not sufficient,
- QC analyses for either WES, CN or panel seq failed.

The experiments were not randomized. The investigators were blinded to the outcome of the clonality score of the different technologies used. Data distribution was not assumed to be normal and appropriate nonparametric statistical tests were used.

**Samples.** Cases of pure primary DCIS that, after treatment, had subsequently developed recurrent disease were identified from:

(1) the Sloane project, a national audit of women with noninvasive neoplasia within the UK National Health Service Breast Screening Programme (REF 08/S0703/147, 19/LO/0648), median follow up 5.3 years[26].
(2) the Dutch DCIS cohort study, a nationwide, population-based patient cohort derived from the Netherlands Cancer Registry (NCR), in which all women diagnosed with primary DCIS between 1989 and 2004 were included, with a median follow up time of 12 years[27]. This cohort was linked to the nationwide network and registry of histology and cytopathology in the Netherlands (PALGA). The study was approved by the review boards of the NCR (reference no. 12.281) and PALGA (reference no. LZV990) and the IRB of the Netherlands Cancer Institute under numbers CFMPB166, CFMPB393 and CFMPB688.
(3) the Duke Hospital cohort, a hospital-based study of women (age 40–75 years) diagnosed with DCIS between 1998 and 2016, with a median follow up of 7.9 years (IRB approvals: Pro00054877 and Pro00068646).

Formalin-fixed paraffin-embedded (FFPE) tissue specimens of patient-matched DCIS and subsequent recurrence were retrieved and reviewed by specialist breast pathologists to confirm the diagnosis and exclude confounding features (such as microinvasion).

In total, 129 DCIS recurrence pairs were included in this study, 95 had developed an ipsilateral invasive recurrence and 34 had an ipsilateral DCIS recurrence. In addition, 34 synchronous DCIS-IBC lesions identified from the Duke DCIS cohort and 14 with a subsequent invasive recurrence in the contralateral breast from the Sloane cohort were also included for testing of the Breakclone algorithm. Details of the cohorts can be found in Supplementary Data Table 1. Associations between clinical variables and clonality were assessed by Fisher's exact test.

**Person-year analysis for invasive breast cancer risk.** We performed person-year analyses to compare the risks of ipsilateral and contralateral invasive breast cancer in the full Dutch cohort from which the Dutch cases and controls were derived with those in the general Dutch female population, overall and for the different treatment groups, allowing one of both or both ipsilateral invasive breast cancer (iiBC) and contralateral invasive breast cancer (ciBC). The time at risk started at date of diagnoses and ended at the date of ipsilateral or contralateral invasive breast cancer, the end of follow up (31 December 2010) or date of death, whichever occurred first. The full Dutch cohort comprised 10,090 primary DCIS patients diagnosed in the period 1989 until 2005 and followed until 1 January 2011. To enable direct comparison of the cumulative incidence rates of iiBC and ciBC, separately and combined, in the full Dutch cohort with death as competing risk, we calculated the expected cumulative incidence in the Dutch general female population.

**DNA isolation.** For DNA isolation, either macrodissection using a light microscope or laser microdissection (LMD) was performed. Sections (8 μm) were stained using nuclear fast red (macrodissection) or toluidine blue (LMD) and DCIS or invasive disease were separated from the normal tissue. Tumor DNA was extracted using the AllPrep DNA/RNA FFPE Kit (Qiagen).

**Exome sequencing.** WES of the paired DCIS with subsequent recurrence together with matched normal tissue was performed at the Department of Genomic Medicine, MD Anderson Cancer Center. Genomic DNA (18–300 ng) was used to generate sequencing libraries using the SureSelectXT Low Input library kit. Libraries were sequenced on NovaSeq 6000, multiplexing 16 tumor samples per lane.

WES of the Duke Hospital Cohort of 34 synchronous paired DCIS and invasive disease and matched normal tissue was performed at the McDonnell Genome Institute at Washington University School of Medicine. Genomic DNA (30–150 ng) was sheared to a mean fragment length of 250 bp and Illumina sequencing libraries were generated as dual-indexed, with unique barcode identifiers, using the Swift Biosciences library kit. Libraries were sequenced on Illumina HiSeq2500 1T instrument by multiplexing nine tumor samples per lane.

Data were converted to a FASTQ format and then aligned to the hg19 reference genome using the Burroughs-Wheeler Aligner (BWA). The aligned BAM files were subjected to mark duplication, realignment and recalibration using Picard v.2.21.9 and GATK v.4.1.7.0. The BAM files were then analyzed by MuTect and Pindel against the matched normal sample to detect somatic single nucleotide variants and insertions/deletions, respectively.

Individuals with normal sample median target coverage (MTC) >40x and tumor sample MTC >80x were included for further investigation. Variants were filtered by the following criteria: (1) Log odds score ≥10; (2) exonic variants; (3) tumor sample coverage at this site ≥15; (4) normal sample coverage at this site ≥10; (5) allele fraction in tumor sample ≥0.02; (6) allele fraction in normal sample <0.01; (7) population frequency <0.01 in ExAC, ESP6500 and 1000G database; (8) hotspot mutations in *PIK3Ca* and *TP53* were added back to the dataset, if they did not pass these criteria and (9) for nonclonal pairs, private mutations were checked manually in the Integrative Genomics Viewer in both the primary and recurrence to ensure that they were indeed private and not filtered out by QC criteria.

We identified potential sample mismatches using an inhouse script for computing SNP matching index. Indexed BAM files from both tumor-normal pairs were used as an input to the variant caller Platypus v.0.5.2 to identify germline variants. For any pair of Platypus vcfs (two samples), we removed the SNPs from random chromosomes as well as SNPs with coverage <10, and calculated the number (nAB) of overlapping SNPs (by position), and the number (nGAB) of the same alleles within the overlapping SNPs. The score (match-index %) = nGAB × 100/nAB. Using this index, we removed all mismatches with score <90%. The details of all mutations detected can be found in Supplementary Data File 1.

**Copy number analysis.** Somatic copy number aberrations (SCNAs) were ascertained using the HumanCytoSNP-12 BeadChip Kit (Illumina) in cases with 100–250 ng of DNA available from the Sloane Project. DNA was restored with the Infinium HD FFPE DNA Restore Kit (Illumina). Raw SNP array data was processed with GenomeStudio 2.0 software (Illumina) and subsequently with the ASCAT (allele-specific copy number analysis of tumors) software algorithm (implemented in R), to estimate allele-specific copy number, the aberrant cell fraction and tumor ploidy[28]. Copy number profiles with number of segments higher than 500 and a log R ratio (LRR) noise higher than 0.16 were removed from the analysis. PLINK v.1.07 was used to estimate the pairwise relatedness using the raw SNP genotyping data to exclude sample mismatches between paired primary DCIS and recurrences.

In cases with limited DNA, copy number was ascertained using low-pass whole-genome analysis. The UK cohort used NEBNext Ultra II DNA Library Prep Kit for Illumina as per manufacturer's instructions and the Dutch cohort used the KAPA hyper prep kit (KAPA Biosystems), protocol KR0961-v.5.16. Agilent S5XT-2 (1–96) adapters with Illumina P5 and P7 sequences were used, containing 8 bp Agilent indices.

Libraries were pooled and sequenced single-end on a HiSeq2500 sequencer (Illumina). After demultiplexing, FASTQ files were aligned to the human reference genome GRCh38 (hg38) using BWA v.0.7.17 aligner and converted to BAM files with SAMtools v.1.9. Duplicate reads were marked with Picard v.2.18.3 and removed, together with reads with mapping qualities lower than 37 using SAMtools v.1.9. Samples were sequenced at an average of 0.2× and minimum 0.03× for the Dutch cohort and average of 0.3× and minimum 0.04× for the UK cohort. Relative copy number profiles were obtained with QDNAseq v.1.22.0 after setting a 100 kb fixed bin size. A bin size of 100 kb was used as this gave the best balance between sensitivity and noise in our data experience. We filtered out copy number profiles with a number of segments >400 and observed/expected noise ratio >50. We used CGHcall v.2.48.0 for relative copy number calling. We filtered out profiles that did not show any copy number aberrations. Despite these QC criteria, there were still a small subset of samples with poor quality copy number profiles, which could lead to incorrect copy number calls and the potential to erroneously call a pair independent if one pair of a sample was of poorer quality than the other. All copy number plots were therefore assessed visually and independently by two experts (L.F.A.W. and E.H.L.) and those that were considered by both to be of poor quality such that copy number calling may not be accurate were excluded. Overall, 14 primaries and four recurrences out of 159 pairs failed the visual inspection.

For detecting differential copy number variation between groups, absolute copy number calls and Fisher's exact test were used. In the samples processed by SNP genotyping, absolute copy number calls were determined relative to tumor ploidy. If the copy number of a segment was more than 0.6 above tumor ploidy, it was called as a gain. If the copy number of a segment was more than 0.6 below tumor ploidy, the SNP was called as a loss. Due to the noisy LRR profiles, we filtered out calls with no BAF signal. In low-pass whole-genome sequenced samples, absolute copy

number calls were obtained after tumor cell fraction adjustment with ACE v.1.4.0. The copy number profiles for all pairs can be found in Supplementary Fig. 1.

**Targeted sequencing.** For the UK cohort, sequencing of all exons of a custom 121 breast cancer-associated gene panel (Supplementary Data Table 10) was performed using the SureSelectXT low input Target Enrichment System (Agilent Technologies); 100 bp read paired-end sequencing was performed on the HiSeq2500 platform. The sequencing output was aligned to the reference genome hg19 using the BWA-MEM (maximal exact match). Variants were called using MuTect2 from the Genome Analysis Toolkit (v.4.1.0.0), using the matched normal tissue to exclude germline variants. Variants with an allele frequency <5%, coverage <30× were excluded. Sequencing reads of tumor and normal pairs were visualized on the Integrative Genomics Viewer to exclude germline variants and also potential sequencing artefacts.

The Dutch cohort was sequenced using an IonTorrent AmpliSeq custom 53-gene panel (Supplementary Data Table 11) and were processed according to the Ion AmpliSeq Library Kit Plus protocol (ThermoFisher Scientific). Reads were aligned to the reference genome GRCh37 (hg19) using the Torrent Mapping Alignment Program, and variant calling was performed using Torrent Variant Caller (TVC) v.5.6. Variant data in VCF format was first translated to GRCh38 and annotated using bedtools, Picard (https://broadinstitute.github.io/picard/command-line-overview.html), SAMtools, bcftools and VEP, and further analyzed in R, employing vcfR and tidy verse. True somatic variants, identified via filtering during which low quality variants VAF <10%, coverage <100× and a quality (QUAL) of <1,000), artifacts (found in >90% of samples), and germline variants (more than five cases in GNOMAD and GoNL) were removed. Details regarding the amplicon panel design, performance and filtering QC are provided in Supplementary Note 1.

**Single-cell sequencing.** FFPE samples were deparaffinized using the FFPE Tissue Dissociation Kit from MACS (catalog no. 130-118-052). Nuclear suspensions were prepared from the recovered cell suspensions using a 4,6-diamidino-2-phenylindole (DAPI)-NST lysis buffer (800 ml of NST (146 mM NaCl, 10 mM Tris base at pH 7.8, 1 mM CaCl$_2$, 21 mM MgCl$_2$, 0.05% BSA, 0.2% Nonidet P-40)), 200 ml of 106 mM MgCl$_2$, 10 mg of DAPI). The nuclear suspensions were filtered through a 35 mm mesh and single nuclei were flow sorted (BD FACSMelody) into individual wells of 384-well plates from the aneuploid peak (Supplementary Note 2). After sorting single nuclei, direct tagmentation chemistry was performed following the acoustic cell tagmentation (ACT) protocol[10]. Briefly, nuclei were lysed and tagmentation was performed using TN5 to add dual barcode adapters to the DNA, followed by 12 cycles of PCR. The resulting libraries were QCed for concentration >10 ng μl$^{-1}$ and pooled for sequencing on the HiSeq4000 (Illumina) instrument at 76 cycles.

To calculate single-cell copy number profiles, we demultiplexed sequencing data from each cell into FASTQ files, allowing one mismatch of the 8 bp barcode. FASTQ files were aligned to hg19 (NCBI Build 37) using bowtie2 (v.2.1.0)[29] and converted from SAM to BAM files with SAMtools (v.0.1.16)[30]. PCR duplicates were removed based on start and end positions. Copy number profiles were calculated at 220 kb resolution using the variable binning method[31]. The preprocessing steps to compute DNA copy number profiles have been described in detail previously[9]. Single cells with <10 median reads per bin were excluded for downstream copy number analysis. GC-normalized read counts were binned into bins of variable size, averaging 200 kb, followed by population segmentation with the multipcf[32] (gamma = 10) method from the R Bioconductor multipcf package. The log$_2$ copy number ratio was calculated and used for subsequent analysis. We filtered out noisy single cells with mean nine-nearest neighbor correlation less than 0.85. The mean nine-nearest neighbor correlation is calculated as the average of the Pearson correlation coefficients between any single cell and its nine-nearest neighbors. This step removed single cells with poor whole-genome amplification from the subsequent data analysis. Single-cell ratio data was embedded into two dimensions using UMAP[33], R package 'uwot' (v.0.1.8, seed = 31, min dist = 0.2, n_neighbors = 30, distance = 'manhattan'). The resulting embedding was used to create an SNN graph with R Bioconductor package scran (v.1.14.6)[34]. Subclones were identified with R package 'dbscan' (v.1.1-5, k_minor = 0.02×no. of cells)[35]. Heatmaps were plotted with R package ComplexHeatmap (v.2.2.0)[36].

**Clonal relatedness calculation using Breakclone.** Breakclone is an inhouse package to assess clonal relatedness. Unlike other packages[12,14], it incorporates both population frequency and allele frequency when using mutation data for determining clonal relatedness. When using copy number data, it uses the position of the individual copy number aberration breakpoints rather than aberration events at the chromosome arm level to determine clonal relatedness correcting for the frequency of the event within the cohort. These are harder to compare across cohorts analyzed with different techniques but, we believe, provide much stronger evidence of clonal relatedness when shared between lesions[37]. A reference distribution of concordance scores is calculated by randomly permuting all possible pairs from different patients, the number of permutations empirically determined as necessary for the distribution to converge and is used to calculate P values for the concordance score of each tumor pair. The threshold for determining clonal relatedness is set as P < 0.01. Clonality scores between 0.05 and 0.01 were called

ambiguous. All values above 0.05 were considered as nonclonal. We considered a sample pair as clonally related if at least one of the different methods (WES, copy number, panelseq) gave a clonal score, with P102 as an exception, as only the copy number data showed borderline relatedness.

**Copy number data.** Each breakpoint shared between two tumors is interpreted as evidence of their relatedness, while each breakpoint unique to one tumor is interpreted as evidence of independence. However, given the generally stochastic process of genomic instability, a common aberration provides stronger evidence than an independent once; therefore, the effect of the unique aberrations in the score calculations is weighted down by one-half.

The concordance score range starts at zero for samples that share no aberrations and approaches the theoretical limit of 1 as the samples become more similar—the score for any two identical samples will be slightly below 1 due to the population frequency corrections.

Each SCNA breakpoint was compared between the pairs of tumors from the same individual. Concordant breakpoints were defined as the same type of aberration, present in the same location ± 5x *average interprobe length* to account for technical variation. This figure was determined empirically as the number that captured the most likely concordant breakpoints without compromising their uniqueness—larger values led to the same breakpoints being included in calculations twice. Each concordant breakpoint was adjusted for its frequency in the entire cohort ($f_b$), producing an adjusted breakpoint concordance score ($s_b$) based on the equation:

$$s_b = 1 - f_b$$

Samples bearing whole-genome duplications (WGD) present a particular challenge in assessing clonal relatedness. Relatedness would be especially underestimated in the case of clonally related samples, only one of which has undergone WGD, as this single event would be interpreted by the algorithm as a large number of gains and amplifications across the entire genome, obscuring the true common events between the two samples. To that end, a correction is applied, by inferring the most likely copy number state that would have existed before the WGD event; for example, an allelically balanced tetraploid region, which would be normally interpreted as a gain, is likely to have originated from a pre-WGD region of normal copy number, and will be corrected accordingly. All relevant corrections applied are presented in the Supplementary Data Table 12. This correction is applied only in the case of SNP array data, which enables the detection of WGD.

The final sample concordance score ($s$) was then calculated between the pairs using all of the SCNAs in the samples and taking into account the total number of breakpoints in both samples ($n_b$), using the following formula:

$$s_s = \frac{\sum s_b}{\sum s_b + \frac{1}{2} \times (n_b - 2 * \sum s_b)}$$

A reference distribution of concordance scores was calculated using all possible tumor pairs from different patients and was used to calculate P values for the concordance score of each tumor pair.

**Somatic mutation data.** The allele frequency is weighted according to the population frequency: the lower the population frequency, the higher the weight of the allele frequency. In the calculations, the square root of the population frequency is used to normalize the range of possible values. The range of values for this score is between 0 for samples with no shared mutations and 1 for samples with identical mutation profiles.

Mutation data from each sample was compared and common variants were assigned a score, based on both their allele frequency in each sample ($A_1$ and $A_2$), and their frequency in the population ($P_c$). A higher allele frequency is interpreted as a stronger indicator of clonal relatedness, while a higher population frequency is interpreted as diminishing the predictive value of the variant. The TCGA Pan-Cancer Atlas breast cancer mutation calls were used for this adjustment, in addition to the mutations found in our cohort. The concordance score ($s_s$) was subsequently calculated, taking into account the private variants in both tumor samples and their allele ($A_p$) and population ($P_p$) frequencies, using the following formula:

$$s_s = \frac{\sum \frac{A_1 + A_2}{\sqrt{P_c}}}{\sum \frac{A_1 + A_2}{\sqrt{P_c}} + 0.5 \times \sum \frac{A_p}{\sqrt{P_p}}}$$

A reference distribution of concordance scores was calculated using all possible tumor pairs from different patients and was used to calculate P values for the concordance score of each tumor pair.

An implementation of the method is available as an R package at github.com/argymeg/breakclone.

Validation of Breakclone in:

(1) Synchronous DCIS and invasive disease

Results from WES on 34 synchronous DCIS-INV pairs were run through breakclone and, as expected, confirmed that most were clonally related (31/34, 91%). The three pairs that were considered unrelated did not share any mutations

and also had the least number of mutations overall, which may suggest that methodological limitations were preventing us from detecting their common origin (Supplementary Data Table 13).

(2) Pure DCIS with contralateral recurrence

A total of 14 pure DCIS and paired contralateral invasive recurrences were analyzed by SNP array and the clonality score calculated by Breakclone. The results showed, as expected, that none of the contralateral invasive recurrences were related to their primary DCIS (Supplementary Data Table 14).

**Comparison of the different clonality algorithms.** In order to assess the added usefulness of our method, we applied the relevant copy number and mutation-based functions in the previously published Clonality package[13,14] to our samples where copy number was ascertained by SNP array (Supplementary Note 3). We observed with the estimate of the number of clonally related pairs was lower for the clonality package compared with our proposed method, as well as suggesting a number of the contralateral recurrences were clonally related. Visual inspection of the contralateral samples that were called clonal showed that they were generally genomically stable samples, with few main aberrations. When those samples do share aberrations on the same chromosomal arms, they are considered clonally related by the Clonality package which, by design, relies on fewer events, but not by our method, which relies on the presence of multiple copy number events.

**Reporting summary.** Further information on research design is available in the Nature Research Reporting Summary linked to this article.

## Data availability

Sequence data has been deposited at the European Genome-phenome Archive (EGA), which is hosted by the EBI and the CRG, under accession number EGAS00001005784 and can be accessed once a data sharing agreement had been signed. For access please contact: elinor.sawyer@kcl.ac.uk and j.wesseling@nki.nl. Further information about EGA can be found on https://ega-archive.org 'The European Genome-phenome Archive of human data consented for biomedical research'. A detailed description of the data collection has been provided in Methods. All associated analyses have been reported on github: https://github.com/argymeg/precision-clonality-code. The following datasets were used to generate the data in the manuscript: TCGA Pan-Cancer Atlas breast cancer mutation calls (https://www.cancer.gov/about-nci/organization/ccg/research/structural-genomics/tcga), hg19 (NCBI Build 37) (https://www.ncbi.nlm.nih.gov/assembly/GCF_000001405.13/_) and GRCh38 (hg38) (https://www.ncbi.nlm.nih.gov/assembly/GCF_000001405.26/).

## Code availability

All code used in this study, apart from code relating to single-cell sequencing, can be accessed at: https://github.com/argymeg/precision-clonality-code
Breakclone Scripts https://doi.org/10.5281/zenodo.6392019
WES Scripts https://doi.org/10.5281/zenodo.6404965
General scripts https://doi.org/10.5281/zenodo.6392150
Single-cell sequencing code can be accessed at https://github.com/navinlabcode/PRECISION_clonality_sc
Single-cell scripts https://doi.org/10.5281/zenodo.6406986

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

## Acknowledgements

Authors listed from ** to ***are in alphabetical order. WES sequencing support was provided by the Moon Shot Cancer Genomics Laboratory, X. Song and L.D. Little, Department of Genomic Medicine, The University of Texas MD Anderson Cancer Center, Houston, TX, USA. We thank all patients in the US, The Netherlands and the UK who have donated their data and tissue for this work. We also wish to thank all the collaborating hospitals, and in particular, pathology departments, and all persons who have helped in the process of data collection and analysis. The authors thank the registration team of the Netherlands Comprehensive Cancer Organization (IKNL) for the collection of data for the Netherlands Cancer Registry. We thank PALGA, the nationwide network and registry of histo- and cytopathology in the Netherlands, for providing pathology data and for their help in the collection of the residual patient material. We acknowledge the staff of the NKI-AVL Core Facility Molecular Pathology and Biobanking for their technical support and the staff of the NKI-AVL Genomics Core Facility and Ronald van Marion of the Erasmus University Medical Center for their sequencing support. We thank T. Hardiman and R. Kataria for their input during development of the Breakclone method. The data for the Sloane Project is based on information collected and quality assured by the PHE Population Screening Programmes. Access to the data was facilitated by the PHE Office for Data Release. We are also grateful for T. Lynch, who helped to expertly coordinate the PRECISON team at Duke University. This work was supported by Cancer Research UK and KWF Kankerbestrijding (reference C38317/A24043) and Breast Cancer Now (2014NovTAP379) with support from Walk the Walk. E.J.S. is funded by a Career Development and Innovation Cancer Award, Guys and St Thomas' Charity and the Cancer Research UK King's Health Partners Centre at King's College London. T.K. is funded by a T32 Translational Genomics Fellowship. This work was supported by grants to N.E.N. by the National Cancer Institute (RO1CA240526) and the CPRIT Single Cell Genomics Center (RP180684). E.S.H. is funded by RFA-CA-17-035 (NIH), 1505-30497 (PCORI), BCRF 19-074 (BCRF), DOD BC132057 and R01 CA185138-01. S.N.Z. is funded by a CRUK Advanced Clinician Scientist Fellowship (C60100/A23916) and supported by the NIHR Cambridge BRC (BRC-125-20014). A.G. is funded by BCN KCL-Q3. A.M.S. is funded by Birmingham CRUK Centre (C17422/A25154). J.K., L.K., T.H., A.F., D.M., C.M. and T.H. are funded by RFA-CA-17-035 (NIH; Hwang). This paper represents independent research part funded by the National Institute for Health Research (NIHR) Biomedical Research Centre at Guy's and St Thomas' NHS Foundation Trust and King's College London, supported via its BRC Genomics Research Platform. The views expressed are those of the authors and not necessarily those of the NHS, the NIHR or the Department of Health and Social Care.

## Author contributions

E.H.L., L.F.A.W., E.S.H., N.E.N., P.A.F., A.M.T., J.W. and E.J.S. designed and supervised the study, interpreted the data and prepared manuscript. T.K. analyzed and interpreted the data and prepared the manuscript. A.M. developed methodology, analyzed and interpreted the data and prepared the manuscript. L.L.V. was involved in study design, data collection, data analysis, data interpretation and preparation of the manuscript. M. Sheinman, P.A.F., V.S., M.H., D.M., M.R.-E., A.A.A., L.M.K. and C.S.-S. collected, analyzed and interpreted data and prepared the manuscript. E.S. provided technical support for single-cell sequencing. M.X. provided technical support for WES. W.M., M.K., P.K., M.d.M., L.M., F.N., R.S., T.M.H. and J.Z. provided technical support. K.C., L.F., B.A.M. and A.M.S. were involved with data collection. S.N., J.Q., M. Sridharan and A.G. provided bioinformatics support. C.C.M. and S.P. designed and supervised the study. H.R.D., J.R.M., S.N.-Z. and C.M. supervised the study and interpreted data. M.Schaapveld and M.K.S. designed the study and interpreted data. A.W.v.d.B.-D. and M.K.S. performed Cancer registry analysis. H.S., D.C. and A.K.C. prepared the manuscript. All authors reviewed the manuscript.

## Competing interests

H.R.D. and S.N.Z. hold patent filings on algorithms for tumor classification (PCT/EP2017/060294PCT/EP2017/060289, PCT/EP2017/060279). The remaining authors declare no competing interests.

## Additional information

**Extended data** is available for this paper at https://doi.org/10.1038/s41588-022-01082-3.

**Correspondence and requests for materials** should be addressed to Elinor J. Sawyer.

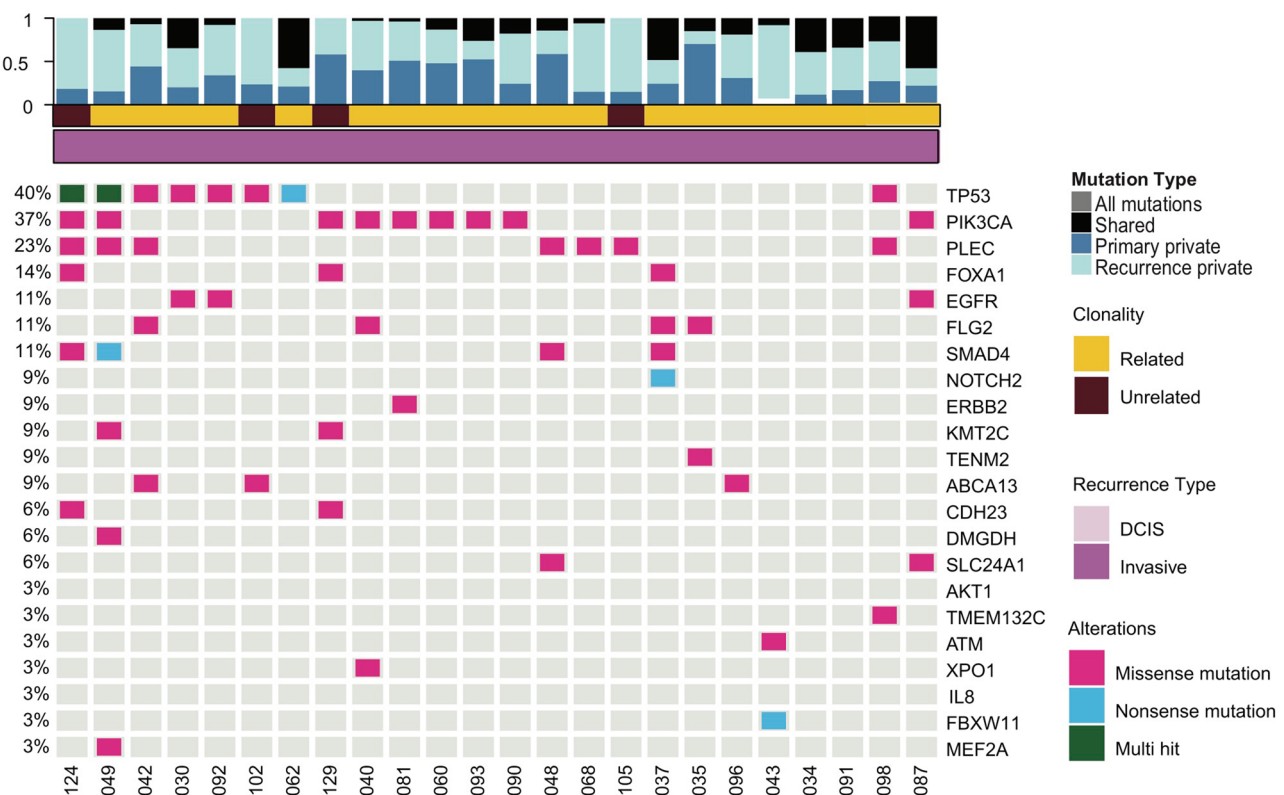

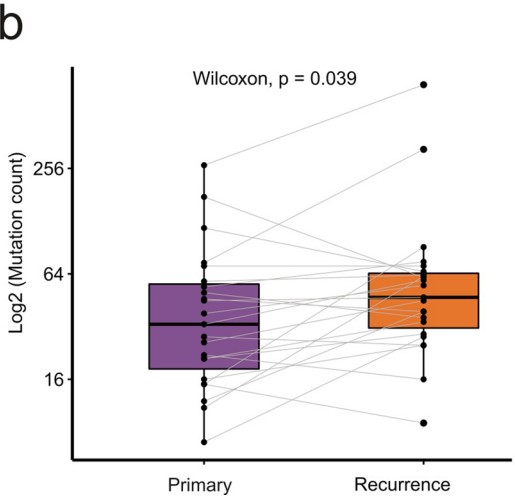

**Extended Data Fig. 1 | Mutational landscape of DCIS samples.** a, Distribution of mutations in subsequent samples with DCIS and invasive recurrences based on WES data (n = 24). b, Boxplot comparing mutation counts in primary DCIS vs invasive recurrences (p = 0.039, n = 23). P value was computed using paired Wilcoxon test. Minima and maxima are present in the lower and upper bounds of the boxplot, respectively. Center solid lines represent the median, box edges show the 25th and 75th percentiles, whiskers represent the maximum and minimum data points within 1.5× interquartile range outside box edges.

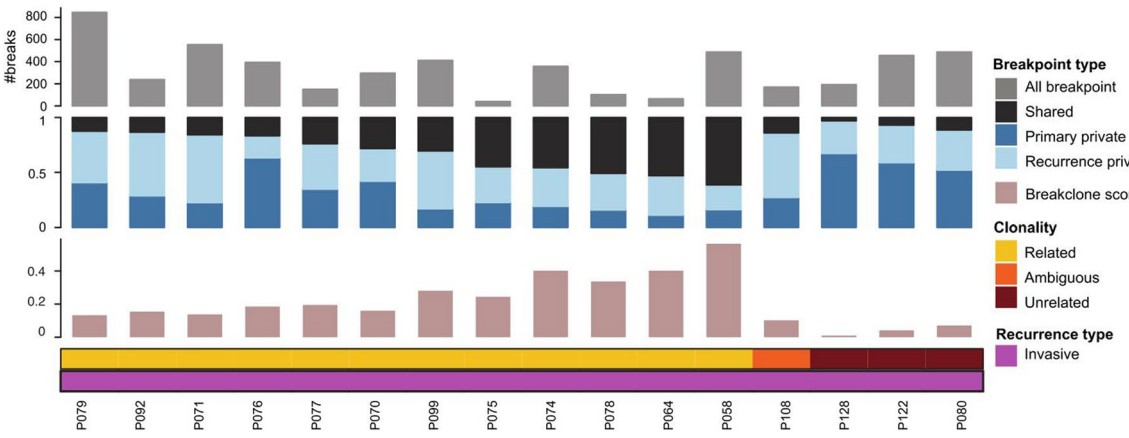

**Extended Data Fig. 2 | Clonality assessment in 16 DCIS-INV pairs where copy number was assessed by SNP-array.** The top row (gray) shows the total number of breakpoints for each patient, the next row whether the breakpoints were unique (private) to the primary or recurrence or shared and the final row (pink) the breakclone score.

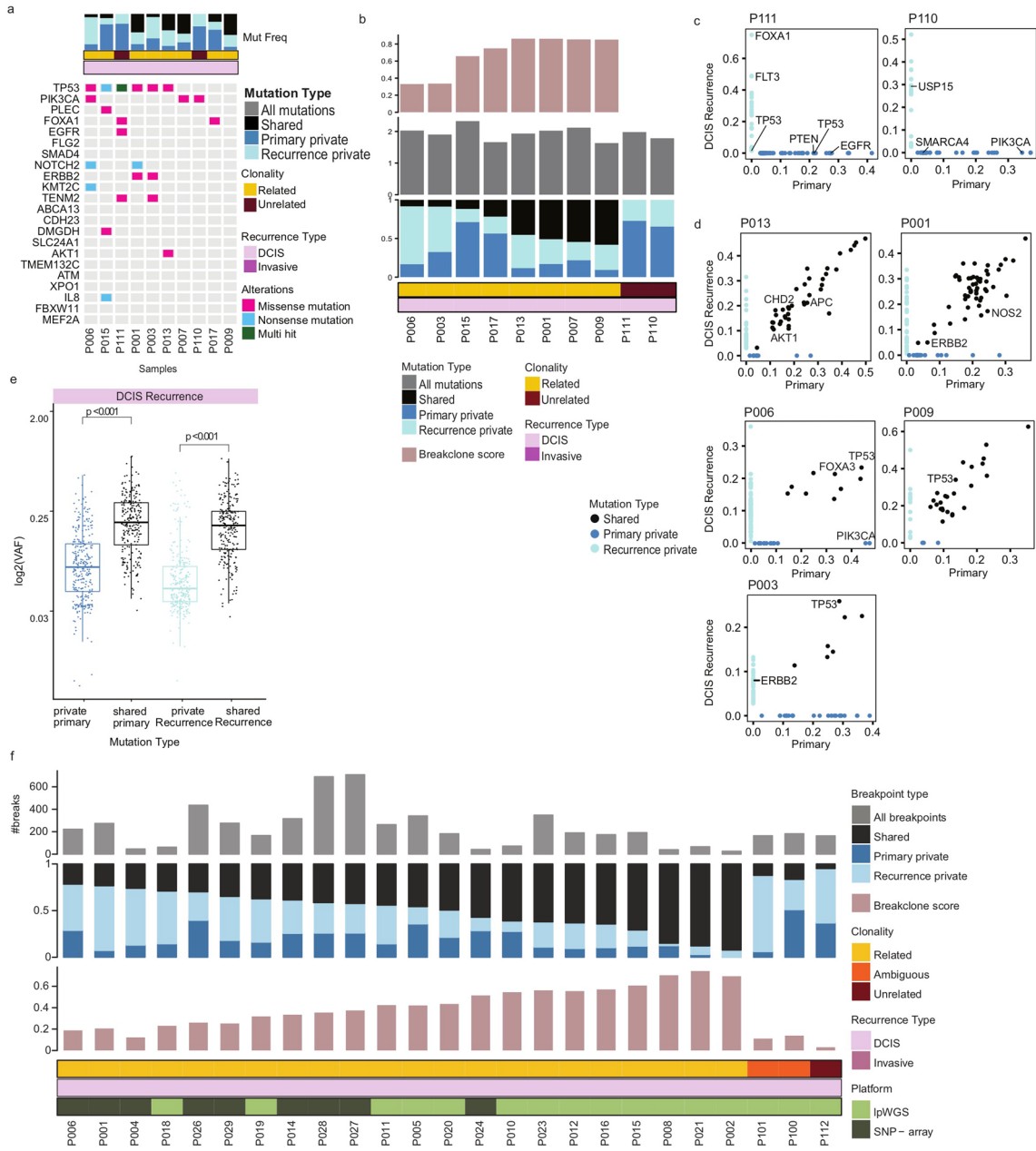

**Extended Data Fig. 3 | Clonality assessment in primary DCIS-DCIS recurrence pairs.** a, Oncoplot of mutations present in the 10 primary DCIS which recurred as DCIS. b, The total number of mutations, followed by the mutations type (primary private, recurrence private and shared), the breakclone score and the final clonality conclusion plotted for 10 DCIS-DCIS recurrence pairs. c, Scatter plots showing the variant allele frequency of mutations in 2 clonally unrelated related pairs. d, Similar as c, for 4 clonally related pairs. e, Boxplots comparing variant allele frequency between private primary (n = 283), private recurrence (n = 289) and shared mutations (n = 241) showing that, as in DCIS-INV pairs, shared mutations had significantly higher allele frequencies compared to private mutations. Minima and maxima are present in the lower and upper bounds of the boxplot, respectively. Adjusted p-values for Holm–Bonferroni method p = 5.20 ×10−48 and p = 1.84 ×10−35, were calculated with two-sided Wilcoxon test. For the shared mutations, both the variant allele frequency in the DCIS tissue (Primary shared) and the DCIS recurrence (Recurrence shared) are shown. Center solid lines represent the median, box edges show the 25th and 75th percentiles, whiskers represent the maximum and minimum data points within 1.5× interquartile range outside box edges. f, Distribution of breakpoints in 25 primary DCIS and recurrent DCIS pairs derived from copy number. The top row (gray) shows the total number of breakpoints for each patient, the next row whether the breakpoints were unique (private) to the primary or recurrence or shared and the final row (pink) the breakclone score.

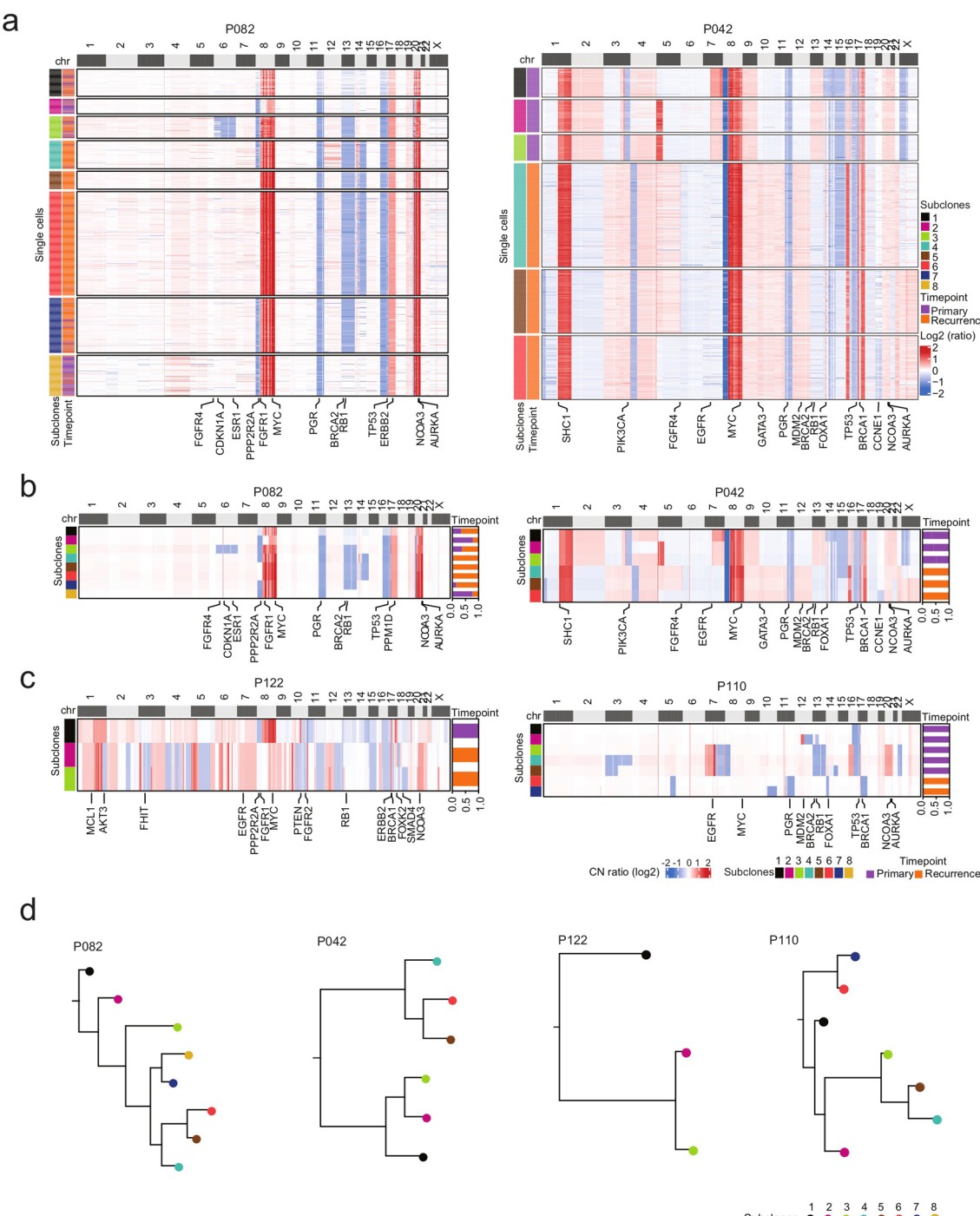

**Extended Data Fig. 4 | Additional clonal lineages inferred from single-cell genome sequencing.** a, Clustered heatmaps of single-cell copy number profiles in genomic order from two DCIS cases with related clonal lineages, with cluster and timepoint information on the right panels and selected breast cancer genes annotated below. b-c, Consensus copy number heatmaps of subclones calculated from clusters of single-cell copy number profiles from clonally related (b,) and clonally unrelated (c,) patients. d, Neighbor-joining trees of clonal lineages constructed from consensus subclones from clonally related and unrelated patients rooted by a diploid node.

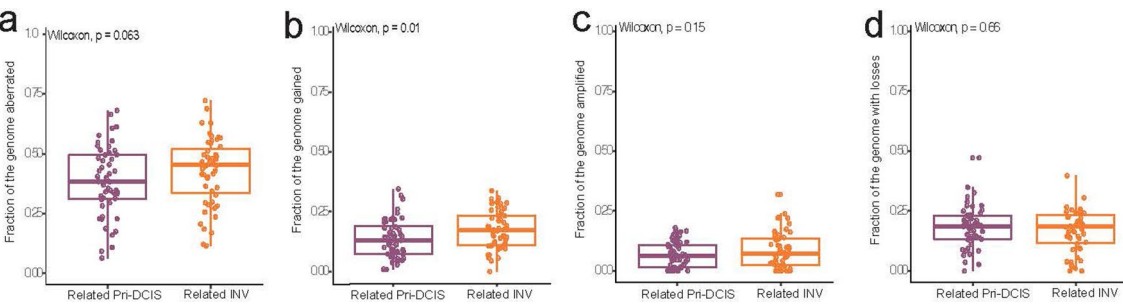

**Extended Data Fig. 5 | Fraction of genome aberrated assessed by copy number in clonally related primary DCIS and invasive recurrence pairs (n = 55).** The boxplots present the distribution of a, Total fraction of genome aberrated; b, Fraction of genome gained; c, Fraction of genome amplified; d, Fraction of genome lost for all the primary DCIS cases (purple) and the invasive recurrences (orange). One-sided Wilcoxon signed-rank test p-values are shown and reveal that invasive recurrences have more copy number gains than the primary DCIS. Minima and maxima are present in the lower and upper bounds of the boxplots, respectively. Center solid lines represent the median, box edges show the 25th and 75th percentiles, whiskers represent the maximum and minimum data points within 1.5× interquartile range outside box edges.

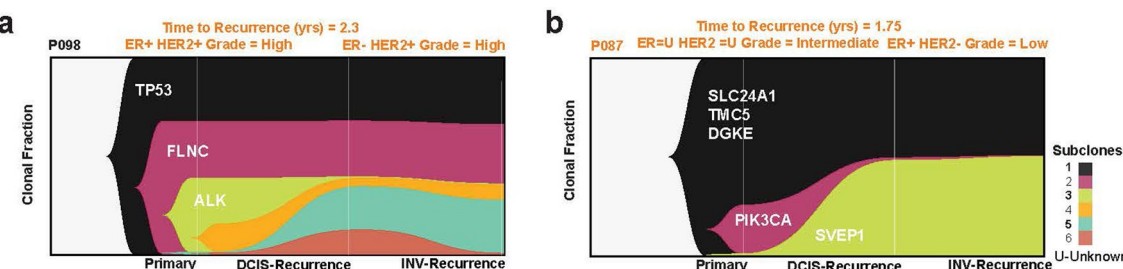

**Extended Data Fig. 6 | Lineage tracing for two cases which recurred as invasive disease with co-existing DCIS and WES was performed on primary DCIS, recurrent DCIS and recurrent invasive disease.** Mueller plots showing clonal frequencies and lineages reconstructed from neighbor-joining trees using timescape. a, recurrent invasive disease comprised four subclones, two of which were detected in the initial primary DCIS and two that appeared in the recurrent synchronous DCIS and invasive disease; b, recurrent invasive disease comprised two subclones one of which the major subclone present in the primary DCIS and the second emerged in the recurrent DCIS and invasive disease. The second subclone present in the primary DCIS which contained a PIK3CA mutation was not found in the recurrent disease.

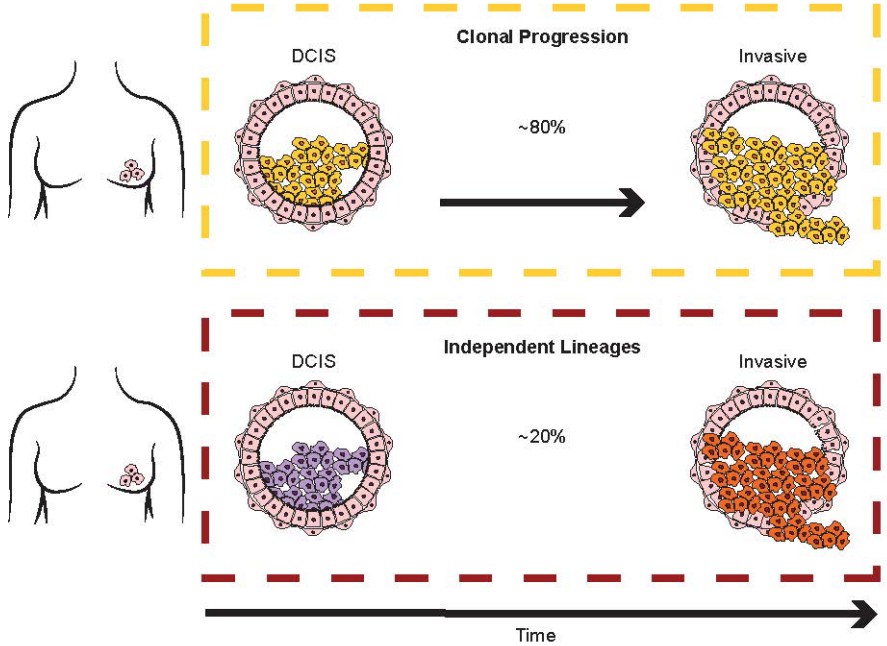

**Extended Data Fig. 7 | Two models for matched DCIS-invasive breast cancer recurrences.** Here we present two models for invasive breast cancer recurrence years after the diagnosis of a primary pure DCIS. a, In 80% of the primary DCIS-matched invasive recurrences a clonal relationship is seen, that is similar mutations or copy number profiles are detected in both lesions. Here, DCIS is a true cancer precursor. b, In 20% of the primary DCIS-matched invasive recurrence no clonal relationship is observed, indicating independent lineages and the likelihood of a second primary cancer. In these lesions DCIS is a risk lesion for an invasive cancer and not a precursor lesion.

| | |
|---|---|

# Reporting Summary

## Statistics

For all statistical analyses, confirm that the following items are present in the figure legend, table legend, main text, or Methods section.

| n/a | Confirmed | |
|---|---|---|
| ☐ | ☒ | The exact sample size (*n*) for each experimental group/condition, given as a discrete number and unit of measurement |
| ☐ | ☒ | A statement on whether measurements were taken from distinct samples or whether the same sample was measured repeatedly |
| ☐ | ☒ | The statistical test(s) used AND whether they are one- or two-sided<br>*Only common tests should be described solely by name; describe more complex techniques in the Methods section.* |
| ☐ | ☒ | A description of all covariates tested |
| ☐ | ☒ | A description of any assumptions or corrections, such as tests of normality and adjustment for multiple comparisons |
| ☐ | ☒ | A full description of the statistical parameters including central tendency (e.g. means) or other basic estimates (e.g. regression coefficient) AND variation (e.g. standard deviation) or associated estimates of uncertainty (e.g. confidence intervals) |
| ☐ | ☒ | For null hypothesis testing, the test statistic (e.g. *F*, *t*, *r*) with confidence intervals, effect sizes, degrees of freedom and *P* value noted<br>*Give P values as exact values whenever suitable.* |
| ☒ | ☐ | For Bayesian analysis, information on the choice of priors and Markov chain Monte Carlo settings |
| ☒ | ☐ | For hierarchical and complex designs, identification of the appropriate level for tests and full reporting of outcomes |
| ☐ | ☒ | Estimates of effect sizes (e.g. Cohen's *d*, Pearson's *r*), indicating how they were calculated |

*Our web collection on statistics for biologists contains articles on many of the points above.*

## Software and code

Policy information about availability of computer code

| | |
|---|---|
| Data collection | A detailed description of the data collection has been provided in the methods section. All associated analyses have been reported on github: https://github.com/argymeg/precision-clonality-code.<br>Single cel sequencing code can be found at: https://github.com/navinlabcode/PRECISION_clonality_sc<br>The following datasets were used to generate the data in the manuscript:<br>TCGA Pan-Cancer Atlas breast cancer mutation calls (https://www.cancer.gov/about-nci/organization/ccg/research/structural-genomics/tcga)<br>hg19 (NCBI Build 37) (https://www.ncbi.nlm.nih.gov/assembly/GCF_000001405.13/_)<br>GRCh38 (hg38) (https://www.ncbi.nlm.nih.gov/assembly/GCF_000001405.26/) |
| Data analysis | A detailed description of the data analyses methods has been provided in the methods section. All associated analyses have been reported on github: https://github.com/argymeg/precision-clonality-code<br>The following data analysis packages were used to generate the data in the manuscript: PLINK v1.07, BWA 0.7.17 , Samtools 1.9, Picard 2.18.3, QDNAseq 1.22.0, CGHcall 2.48.0, BWA-MEM, MuTect2 from the Genome Analysis Toolkit (version 4.1.0.0), Torrent Variant Caller (TVC) version 5.6, bcftools, VEP, SAMtools (0.1.16), UMAP, bowtie2 (2.1.0) , R Bioconductor multipcf package, R package 'uwot' v0.1.8, R Bioconductor package scran (v1.14.6), R package 'dbscan' v1.1-5, R package ComplexHeatmap v2.2.0, Breakclone, Clonality package. |

For manuscripts utilizing custom algorithms or software that are central to the research but not yet described in published literature, software must be made available to editors and reviewers. We strongly encourage code deposition in a community repository (e.g. GitHub). See the Nature Portfolio guidelines for submitting code & software for further information.

March 2021

## Data

Policy information about availability of data

All manuscripts must include a data availability statement. This statement should provide the following information, where applicable:

- Accession codes, unique identifiers, or web links for publicly available datasets
- A description of any restrictions on data availability
- For clinical datasets or third party data, please ensure that the statement adheres to our policy

Sequence data has been deposited at the European Genome-phenome Archive (EGA), which is hosted by the EBI and the CRG, under accession number EGAS00001005784 and can be accessed once a data sharing agreement had been signed. Contact details for the Access Committee are available on the EGA webpage.

Further information about EGA can be found on https://ega-archive.org "The European Genome-phenome Archive of human data consented for biomedical research"( http://www.nature.com/ng/journal/v47/n7/full/ng.3312.html ).

A detailed description of the data collection has been provided in the methods section. All associated analyses have been reported on github: https://github.com/argymeg/precision-clonality-code

The following datasets were used to generate the data in the manuscript:

TCGA Pan-Cancer Atlas breast cancer mutation calls (https://www.cancer.gov/about-nci/organization/ccg/research/structural-genomics/tcga)

hg19 (NCBI Build 37) (https://www.ncbi.nlm.nih.gov/assembly/GCF_000001405.13/_)

GRCh38 (hg38) (https://www.ncbi.nlm.nih.gov/assembly/GCF_000001405.26/)

# Field-specific reporting

Please select the one below that is the best fit for your research. If you are not sure, read the appropriate sections before making your selection.

☒ Life sciences          ☐ Behavioural & social sciences          ☐ Ecological, evolutionary & environmental sciences

For a reference copy of the document with all sections, see nature.com/documents/nr-reporting-summary-flat.pdf

# Life sciences study design

All studies must disclose on these points even when the disclosure is negative.

| | |
|---|---|
| Sample size | The research question we addressed ('What percentage of primary DCIS is clonally related to invasive recurrences?') has never been addressed on a large scale before. Therefore, we collected all available paired samples from three large cohorts. As invasive recurrences after DCIS are rare, long follow up time is needed, old FFPE materials have poor DNA quality, and the amount of DCIS tissue can be limited, it was very difficult to find suitable sample pairs for deep genomic characterization. By combining three international cohorts, we have the largest sample series to date and the best available sample series to answer our research question. |
| Data exclusions | Samples were excluded due to the following reasons:<br>- Only one sample of a pair was available<br>- DNA quantity was not sufficient<br>- QC analyses for either WES, CN or panel seq failed |
| Replication | We combined four different methods to assess clonality: WES, panel seq, CN analyses and single cell sequencing. All methods showed high concordance, and we are therefore convinced that our data is robust. |
| Randomization | Randomization was not applied here, as there was no intervention in this study. Outcome was already known and all samples were genomically profiled to assess clonality. |
| Blinding | Blinding was not necessary for data collection as we only collected samples with the same outcome - all had recurred). The lab technicians and the data analysts were blinded to the clonality score of the different technologies used. |

# Reporting for specific materials, systems and methods

We require information from authors about some types of materials, experimental systems and methods used in many studies. Here, indicate whether each material, system or method listed is relevant to your study. If you are not sure if a list item applies to your research, read the appropriate section before selecting a response.

## Materials & experimental systems

| n/a | Involved in the study |
|---|---|
| ☒ | ☐ Antibodies |
| ☒ | ☐ Eukaryotic cell lines |
| ☒ | ☐ Palaeontology and archaeology |
| ☒ | ☐ Animals and other organisms |
| ☐ | ☒ Human research participants |
| ☐ | ☒ Clinical data |
| ☒ | ☐ Dual use research of concern |

## Methods

| n/a | Involved in the study |
|---|---|
| ☒ | ☐ ChIP-seq |
| ☐ | ☒ Flow cytometry |
| ☒ | ☐ MRI-based neuroimaging |

# Human research participants

Policy information about studies involving human research participants

| | |
|---|---|
| Population characteristics | Women with pure DCIS who developed a subsequent invasive or DCIS recurrence |
| Recruitment | The research participants were recruited through three studies:<br><br>The Sloane project, United Kingdom National Health Service Breast Screening Programme, was approved by the UK Health Research Authority (Ethical approval REF 08/S0703/147, 19/LO/0648)<br><br>The Dutch DCIS cohort study, a Netherlands Cancer Registry (NCR; ref. no 12.281), nationwide network and registry of histology and cytopathology in the Netherlands (PALGA; ref. no. LZV990) approved by the Central Committee on Research Involving Human Subjects in the Netherlands and the Institutional Review Board of the Netherlands Cancer Institute (CFMPB166, CFMPB393 and CFMPB688).<br><br>The Duke Hospital cohort approved by Duke University Health System Institutional Review Board, USA (Pro00054877, Pro00068646).<br><br>There was no participant compensation. |
| Ethics oversight | UK Health Research Authority<br>The Central Committee on Research Involving Human Subjects of the Netherlands<br>Institutional Review Board of the Netherlands Cancer Institute<br>Duke University Health System Institutional Review Board, USA |

Note that full information on the approval of the study protocol must also be provided in the manuscript.

# Clinical data

Policy information about clinical studies

All manuscripts should comply with the ICMJE guidelines for publication of clinical research and a completed CONSORT checklist must be included with all submissions.

| | |
|---|---|
| Clinical trial registration | The Sloane project, United Kingdom National Health Service Breast Screening Programme (REF 08/S0703/147, 19/LO/0648)<br><br>The Dutch DCIS cohort study, a Netherlands Cancer Registry (NCR; ref. no 12.281), nationwide network and registry of histology and cytopathology in the Netherlands (PALGA; ref. no. LZV990). IRB of the Netherlands Cancer Institute (CFMPB166, CFMPB393 and CFMPB688)<br><br>The Duke Hospital cohort (IRB approvals: Pro00054877, Pro00068646) |
| Study protocol | These datasets are not part of clinical trials. Study protocols are available from the corresponding author. |
| Data collection | The Sloane project, a national audit of women with non-invasive neoplasia within the United Kingdom National Health Service Breast Screening Programme. Data collected between 2003 and 2012.<br><br>The Dutch DCIS cohort study, a nation-wide, population-based patient cohort derived from the Netherlands Cancer Registry (NCR), in which all women diagnosed with primary DICS between 1989 and 2004 were included.<br><br>The Duke Hospital cohort, a hospital-based study of women (age 40-75 years) diagnosed with DCIS between 1998 and 2016. |
| Outcomes | Outcome data of our clinical cohorts consisted of any ipsilateral recurrence, i.e. an in situ or an invasive recurrence; at least 5 months after the diagnosis of a primary DCIS lesion. |

# Flow Cytometry

## Plots

Confirm that:

☐ The axis labels state the marker and fluorochrome used (e.g. CD4-FITC).

☐ The axis scales are clearly visible. Include numbers along axes only for bottom left plot of group (a 'group' is an analysis of identical markers).

☐ All plots are contour plots with outliers or pseudocolor plots.

☐ A numerical value for number of cells or percentage (with statistics) is provided.

## Methodology

| | |
|---|---|
| Sample preparation | FFPE Tissue Dissociation Kit from MACS (Cat#130-118-052) |
| Instrument | BD FACSMelody |
| Software | BD FACSChorus software |
| Cell population abundance | DAPI counts vs DAPI area plots were used to determine relative population abundance |
| Gating strategy | DAPI (area) fluorescence intensities were used to determine which nuclei populations were flow-sorted. We gate from 2N peaks and from > 2N peaks to enrich tumor cells based on DAPI fluorescent signal. |

☒ Tick this box to confirm that a figure exemplifying the gating strategy is provided in the Supplementary Information.

