## [Peer Review File · Nature Genetics]

Peer Review Information

Manuscript Title: Genomic analysis defines clonal relationships of ductal carcinoma in situ and recurrent invasive breast cancer

Corresponding author name(s): Professor Elinor Sawyer

Editorial Notes:

Transferred manuscripts This manuscript has been previously reviewed at another journal. This document only contains reviewer comments, rebuttal and decision letters for versions considered at Nature Genetics.

Reviewer Comments & Decisions:

Decision Letter, initial version:

14th Oct 2021

Dear Professor Sawyer,

Your Article, "Clonal relationships of ductal carcinoma in situ and recurrent invasive breast cancers defined by genomic analysis" has now been seen by 3 referees. You will see from their comments below that while they find the work to be improved, some important points are raised. We are interested in publishing your study in Nature Genetics, but would like you to address a couple of remaining points before we move forward. Please be assured that these points are relatively minor and do not require any further analysis.

The reviewers consider the work to be improved and technically robust. However, both Reviewers #1 and #2 maintain that your work is largely confirming (rather than disputing) the expectations of the community regarding the clonality between DCIS and IDC. As such, they suggest that you reposition some of your narrative to make this clear. From our point of view, the value provided by the paper is the robust demonstration of this using a dataset that is uniquely positioned to address this important question. As such, I think that their requests are reasonable, and editing the text as per their suggestions will not reduce the overall impact of the piece.

The second request is around data deposition which, as you likely know, is an absolute priority for Nature Genetics. I appreciate that you have indicated your intention to deposit the data and I would ask that your revised manuscript and cover letter provide a full data availability statement (including live accession codes).

We therefore invite you to revise your manuscript taking into account all reviewer and editor comments. Please highlight all changes in the manuscript text file. At this stage we will need you to upload a copy of the manuscript in MS Word .docx or similar editable format.

*2) If you have not done so already please begin to revise your manuscript so that it conforms to our Article format instructions, available [here](http://www.nature.com/ng/authors/article_types/index.html). Refer also to any guidelines provided in this letter.

[REDACTED]

We hope to receive your revised manuscript within four to eight weeks. If you cannot send it within this time, please let us know.

Nature Genetics is committed to improving transparency in authorship. As part of our efforts in this

direction, we are now requesting that all authors identified as 'corresponding author' on published papers create and link their Open Researcher and Contributor Identifier (ORCID) with their account on the Manuscript Tracking System (MTS), prior to acceptance. ORCID helps the scientific community achieve unambiguous attribution of all scholarly contributions. You can create and link your ORCID from the home page of the MTS by clicking on 'Modify my Springer Nature account'. For more information please visit www.springernature.com/orcid.

Sincerely,

Safia Danovi
Editor
Nature Genetics

Referee expertise:

Referee #1:

Referee #2:

Referee #3:

Reviewers' Comments:

Reviewer #1:

Remarks to the Author:

The authors have suitably responded to my comments and edited the text as requested. The paper is stronger with response to the reviewers comments.

Reviewer #2:

Remarks to the Author:

This revised manuscript addresses several of the concerns raised with the initial submission, although significant issues remain. Chief among these is the general lack of novelty of the principal findings that most ipsilateral breast cancer recurrences are related to the initial primary, and that genomic alterations in recurrent invasive tumors are already present in DCIS. These findings are consistent with currently thinking and existing data in the literature, which the authors do not appear to dispute.

Most of the technical concerns have been addressed satisfactorily by the authors, with a few exceptions that still would benefit from clarification;

1. In 339-340, the authors state that they investigated variants that are shared at high VAF between the DCIS and paired recurrence as putative germline variants, although they state that "they cannot be sure that they are not somatic". If this is indeed the case, the authors should state explicitly in the manuscript that these shared variants are likely a combination of both somatic and germline variants.
2. The authors then go on to say that "all mutations were excluded from the clonality analysis on the basis that they were likely germline". Is it the case that what the authors 'meant' to say was: "all mutations were excluded from the clonality analysis for patients without normal tissue sequencing". and/or "only CNAs were included in clonality analysis of these samples due to ambiguity in somatic versus germline designation of these variants"? The authors should clarify in the manuscript the specific meaning intended. Additionally, the authors also stated that these shared variants "were likely germline". How do the authors know that the mutations are likely germline? Wouldn't these mutations be expected to be a combination of somatic and germline variants? Or perhaps they expect most of them to be germline? This issue/intended meaning would benefit from clarification in the manuscript.
3. Additionally, in the next sentence the authors state that: "Two pathogenic germline BRCA2 mutations were identified...". Is it the case that these variants were identified in the set of samples without normal samples? If true, the authors cannot state that these are germline variants (BRCA2 somatic mutations are not uncommon). More clarification is needed, e.g "Two variants were found in samples that did not have normal sample sequencing that are known pathogenic germline variants in BRCA2).

Reviewer #3:

Remarks to the Author:

Overall the manuscript is improved and a number of my previous comments addressed. I do believe, however, that the authors continue to overstate the importance of these findings, which were predictable based upon what is known for synchronous DCIS-Invasive pairs, and based upon what is known about LCIS-ILC.

I continue to feel the statement that "The finding that one in five ipsilateral invasive cancer following DCIS are not clonally related refutes the prevailing dogma that all ipsilateral invasive breast cancer arises from progression of the initial DCIS" is too strong a statement, and in my view the prevailing dogma is that "most", or the "vast majority" of DCIS-invasive are clonally related. This is basically a single word change, but the authors place too much emphasis on "all". This small wording difference impacts a later statement that "refuting prevailing beliefs regarding the role of DCIS", where if the dogma is "most" or even "the vast majority", then the work presented here supports the prevailing dogma, instead of refuting it. I suggest the manuscript be adjusted to not be based upon an all or none type statement of "the prevailing dogma", which is an expert opinion of these esteemed authors, but nonetheless, an opinion of what they think the "field thinks".

One other comment includes to note that the data has not yet been deposited into a public archive. Given the rules of data release requirements, these data should already be in the EGP or dbGAP and a password provided to the reviewers and the journal. To provide the reviewers a link to a private data repository is not appropriate. It has been my experience it may take months to get data into these repositories, and thus this needs to be done very soon so as to ensure if and when the paper is published, that the data is ready to be released coincident with publication.

Author Rebuttal to Initial comments**Reviewers' Comments:****Reviewer #1:**Remarks to the Author:

The authors have suitably responded to my comments and edited the text as requested. The paper is stronger with response to the reviewers' comments.

Reviewer #2:Remarks to the Author:

This revised manuscript addresses several of the concerns raised with the initial submission, although significant issues remain. Chief among these is the general lack of novelty of the principal findings that most ipsilateral breast cancer recurrences are related to the initial primary, and that genomic alterations in recurrent invasive tumours are already present in DCIS. These findings are consistent with current thinking and existing data in the literature, which the authors do not appear to dispute.

Author's reply

We agree with this reviewer that our findings are in line with published literature. We have made this even more explicit by removing the statement about 'the prevailing dogma'. We would like to refer the reviewer here to the reply to reviewer's 3 comments below. Here we in detail describe the changes we made to better show that our findings are in line with published literature.

Remarks to the Author:

Most of the technical concerns have been addressed satisfactorily by the authors, with a few exceptions that still would benefit from clarification:

1. In 339-340, the authors state that they investigated variants that are shared at high VAF between the DCIS and paired recurrence as putative germline variants, although they state that "they cannot be sure that they are not somatic". If this is indeed the case, the authors should state explicitly in the manuscript that these shared variants are likely a combination of both somatic and germline variants.
2. The authors then go on to say that "all mutations were excluded from the clonality analysis on the basis that they were likely germline". Is it the case that what the authors 'meant' to say was: "all mutations were excluded from the clonality analysis for patients without normal tissue sequencing". and/or "only CNAs were included in clonality analysis of these samples due to ambiguity in somatic versus germline designation of these variants"? The authors should clarify in the manuscript the specific meaning intended. Additionally, the authors also stated that these shared variants "were likely germline". How do the authors know that the mutations are likely germline? Wouldn't these

mutations be expected to be a combination of somatic and germline variants? Or perhaps they expect most of them to be germline? This issue/intended meaning would benefit from clarification in the manuscript.

3. Additionally, in the next sentence the authors state that: "Two pathogenic germline BRCA2 mutations were identified...". Is it the case that these variants were identified in the set of samples without normal samples? If true, the authors cannot state that these are germline variants (BRCA2 somatic mutations are not uncommon). More clarification is needed, e.g. "Two variants were found in samples that did not have normal sample sequencing that are known pathogenic germline variants in BRCA2).

Author's reply

We thank the reviewer for asking further clarification. Therefore, we edited the section on germline mutations (see below) to address the three points made above by reviewer #2 and make this section clearer. We have also added 2 extra columns to supp. table 9, to make it clear if the sample had paired normal or not and if the variant has ever been listed as somatic in COSMIC.

"In the 38 samples that underwent Panelseq without paired normal tissue we also looked for potential germline mutations in the above seven genes. However to minimise the likelihood of including any somatic mutations due to the lack of paired normal, we only included variants which have previously been described as germline pathogenic variants in clinvar and where the VAF was > 30% in both the DCIS and paired recurrence. Two variants were identified that are known pathogenic germline BRCA2 mutations, one of these has previously been described as somatic as well as germline so we cannot be certain it is a germline mutation in our sample. In both of these cases the invasive recurrences were clonally related to the primary DCIS. Another four variants of unknown significance or conflicting pathogenicity were identified which have not previously been described as somatic; in two the invasive recurrences were clonally related, one equivocal and the other unrelated (Supplementary Data Table 9). All potential germline variants were excluded from the clonality analysis."

Reviewer #3:

Remarks to the Author:

Overall, the manuscript is improved and a number of my previous comments have been addressed. I do believe, however, that the authors continue to overstate the importance of these findings, which were predictable based upon what is known for synchronous DCIS-Invasive pairs, and based upon what is known about LCIS-ILC.

I continue to feel the statement that "The finding that one in five ipsilateral invasive cancer following DCIS are not clonally related refutes the prevailing dogma that all ipsilateral invasive breast cancer arises from progression of the initial DCIS" is too strong a statement, and in my view the prevailing dogma is that "most", or the 'vast majority' of DCIS-invasive are clonally related. This is basically a single word change, but the authors place too much emphasis on "all". This small wording difference impacts a later statement that "refuting prevailing beliefs regarding the role of DCIS", where if the dogma is "most" or even "the vast majority", then the work presented here supports the prevailing dogma, instead of

refuting it. I suggest the manuscript be adjusted to not be based upon an all or none type statement of “the prevailing dogma”, which is an expert opinion of these esteemed authors, but nonetheless, an opinion of what they think the ‘field thinks”.

Author’s reply

We would like to thank the reviewer for these comments that we have taken on board by removing the above statements from the manuscript (highlighted in yellow in the paragraphs below).

“The finding that one in five ipsilateral invasive cancer following DCIS are not clonally related ~~refutes the prevailing dogma that all ipsilateral invasive breast cancer arises from progression of the initial DCIS. These data~~ has fundamental biologic implications: first, DCIS can no longer be considered solely as a precursor lesion, but rather also a risk lesion for development of further invasive disease. This is similar to the role that has been ascribed to lobular carcinoma in situ (LCIS) where there is both an increased risk of subsequent ipsilateral and contralateral invasive disease²². Second, the true risk of recurrence from the same population of preinvasive tumor cells has likely been overestimated, thereby confounding the potential benefit of radiotherapy, as radiation likely prevents clonal progression rather than preventing initiation of a new neoplastic process. Third, these data have important implications for accurate identification of predictive biomarkers for invasive progression, since in clonally unrelated DCIS, the notion of biomarkers predictive of invasion is irrelevant. These data may explain why it has been so challenging to identify predictive biomarkers of progression to invasive disease to date^{23,24}, further underscoring the need to characterize DCIS more comprehensively in the context of the stroma in future studies.

Important future directions will include identifying those factors which contribute to dormancy of DCIS cells and their re-activation to establish invasive disease years to decades later, and understanding the role of non-genetic factors, such as the tumor microenvironment, in invasive progression. These biological insights are essential to enable well-informed DCIS treatment decisions which will help avoid overtreatment of low-risk DCIS that likely will never progress, while still providing appropriately aggressive treatment for high-risk DCIS with greatest invasive potential.

In conclusion, we performed extensive genomic characterization of the primary tumor and matched recurrence in one of the largest cohorts of patients treated for DCIS who subsequently developed an ipsilateral invasive cancer. Although the majority of subsequent invasive cancers were clonally related to the primary DCIS, a substantial subset was unrelated to the index DCIS ~~refuting current prevailing beliefs regarding the role of DCIS. Our findings show that DCIS~~ is not only a precursor to invasive cancer, but also a potential marker of a field effect for which surgery may have a limited role. These insights suggest an increasingly important rationale for developing non-surgical approaches to effectively reduce increased breast cancer risk in these patients.”

Remark to the Author:

One other comment includes to note that the data has not yet been deposited into a public archive. Given the rules of data release requirements, these data should already be in the EGP or dbGAP and a password provided to the reviewers and the journal. To provide the reviewers a link to a private data repository is not appropriate. It has been my experience it may take months to get data into these repositories, and thus this needs to be done very soon so as to ensure if and when the paper is published, that the data is ready to be released coincident with publication.

Author's reply

Data is in the process of being uploaded to the European Genome-phenome Archive under accession number EGAS00001005784.

Decision Letter, first revision:

Our ref: NG-A58483R

30th Nov 2021

Dear Dr. Sawyer,

Thank you for submitting your revised manuscript "Clonal relationships of ductal carcinoma in situ and recurrent invasive breast cancers defined by genomic analysis" (NG-A58483R). We have reviewed your revisions in-house and I'm delighted to say that we'll be happy in principle to publish it in Nature Genetics, pending minor revisions to comply with our editorial and formatting guidelines.

Sincerely,

Safia Danovi
Editor
Nature Genetics

Final Decision Letter:

In reply please quote: NG-A58483R1 Sawyer

22nd Apr 2022

Dear Dr. Sawyer,

I am delighted to say that your manuscript "Genomic analysis defines clonal relationships of ductal carcinoma in situ and recurrent invasive breast cancer" has been accepted for publication in an upcoming issue of Nature Genetics.

Your paper will be published online after we receive your corrections and will appear in print in the next available issue. You can find out your date of online publication by contacting the Nature Press Office (press@nature.com) after sending your e-proof corrections. Now is the time to inform your Public Relations or Press Office about your paper, as they might be interested in promoting its publication. This will allow them time to prepare an accurate and satisfactory press release. Include your manuscript tracking number (NG-A58483R1) and the name of the journal, which they will need when they contact our Press Office.

Please note that *Nature Genetics* is a Transformative Journal (TJ). Authors may publish their research with us through the traditional subscription access route or make their paper immediately open access through payment of an article-processing charge (APC). Authors will not be required to

make a final decision about access to their article until it has been accepted. [Find out more about Transformative Journals](https://www.springernature.com/gp/open-research/transformative-journals)

Authors may need to take specific actions to achieve [compliance with funder and institutional open access mandates](https://www.springernature.com/gp/open-research/funding/policy-compliance-faqs). If your research is supported by a funder that requires immediate open access (e.g. according to [Plan S principles](https://www.springernature.com/gp/open-research/plan-s-compliance)) then you should select the gold OA route, and we will direct you to the compliant route where possible. For authors selecting the subscription publication route, the journal's standard licensing terms will need to be accepted, including [self-archiving-and-license-to-publish](https://www.nature.com/nature-portfolio/editorial-policies/self-archiving-and-license-to-publish). Those licensing terms will supersede any other terms that the author or any third party may assert apply to any version of the manuscript.

Please note that Nature Portfolio offers an immediate open access option only for papers that were first submitted after 1 January, 2021.

If you have not already done so, we invite you to upload the step-by-step protocols used in this manuscript to the Protocols Exchange, part of our on-line web resource, natureprotocols.com. If you complete the upload by the time you receive your manuscript proofs, we can insert links in your article that lead directly to the protocol details. Your protocol will be made freely available upon publication of your paper. By participating in natureprotocols.com, you are enabling researchers to more readily reproduce or adapt the methodology you use. [Natureprotocols.com](https://natureprotocols.com) is fully searchable, providing your

protocols and paper with increased utility and visibility. Please submit your protocol to <https://protocolexchange.researchsquare.com/>. After entering your nature.com username and password you will need to enter your manuscript number (NG-A58483R1). Further information can be found at <https://www.nature.com/nature-portfolio/editorial-policies/reporting-standards#protocols>

Sincerely,

Safia Danovi
Editor
Nature Genetics